# Food demand displaced by global refugee migration influences water use in already water stressed countries

Leonardo Bertassello [1], Marc F. Müller [1] ✉, Adam Wiechman[1], Gopal Penny [1,2], Marta Tuninetti [3] & Michèle C. Müller-Itten[4]

Millions of people displaced by conflicts have found refuge in water-scarce countries, where their perceived effect on water availability has shaped local water security discourses. Using an annual global data set, we explain the effects of refugee migrations on the host countries' water stress through the food demand displaced by refugees and the water necessary to produce that food. The water footprint of refugee displacement increased by nearly 75% globally between 2005 and 2016. Although minimal in most countries, implications can be severe in countries already facing severe water stress. For example, refugees may have contributed up to 75 percentage points to water stress in Jordan. While water considerations should not, alone, determine trade and migration policy, we find that small changes to current international food supply flows and refugee resettlement procedures can potentially ease the effect of refugee displacement on water stress in water-vulnerable countries.

Ensuring the availability and sustainable management of water for all is a defining challenge of our time[1]. This is particularly true when recurring droughts collide with rapid demographic change and enduring armed conflicts[2]. Although almost never the sole cause of conventional wars[3,4], water scarcity may act as a risk factor for civil conflicts[5,6] and a possible linkage between climate change and violence[7]. However, armed conflicts also affect water resources by damaging infrastructure and institutions and disrupting prevailing local water uses[8]. Abandonment of irrigated agriculture in southern Syria during the recent civil war caused a near doubling of river flow volumes into downstream Jordan[9], suggesting that the impact of armed conflicts on water resources can propagate beyond borders along international water ways. This effect on water availability is only half of the story, however, because the conflict also caused at least 1.1 million Syrian refugees to flee across the border into Jordan[10], adding pressure to the country's already scarce water resources[11]. By displacing water demand through refugee migration, conflicts can affect water resources beyond political and topographic boundaries.

As of 2021, approximately 80 million people were forcibly displaced by armed conflicts globally, more than 30 million of whom had to migrate internationally as refugees under mandates from the United Nations High Commissioner for Refugees (UNHCR) or the United Nations Relief and Works Agency for Palestine Refugees in the Near East (UNWRA). Forced migrants under a mandate from either agency are here jointly referred to as 'refugees'. The number of displaced refugees has nearly doubled from 12.1 to 23.1 million in the 2005–2016 period–the sharpest increase on record (see Fig. S1 in Supplementary Information, SI). The majority of displaced refugees during that period hail from countries in regions with arid or semi-arid climates (Fig. 1A), and nearly half of all refugees fled four particular countries or territories: Syria, Iraq, Palestine, and Afghanistan (Fig. 1B). The large majority (87%,[12]) of these migrants crossed into neighboring countries that share similar climate conditions and often already face their own substantial water availability challenges. A growing scholarship focuses on the water-security implications of migration in destination countries. Identified mechanisms include overburdened local

[1]Department of Civil and Environmental Engineering and Earth Sciences, University of Notre Dame, Notre Dame, IN, USA. [2]Department of Geography, National University of Singapore, Singapore, Singapore. [3]Department of Environment, Land, and Infrastructure Engineering, Politecnico di Torino, Turin, Italy. [4]Department of Economics, University of Notre Dame, Notre Dame, IN, USA. ✉e-mail: mmuller1@nd.edu

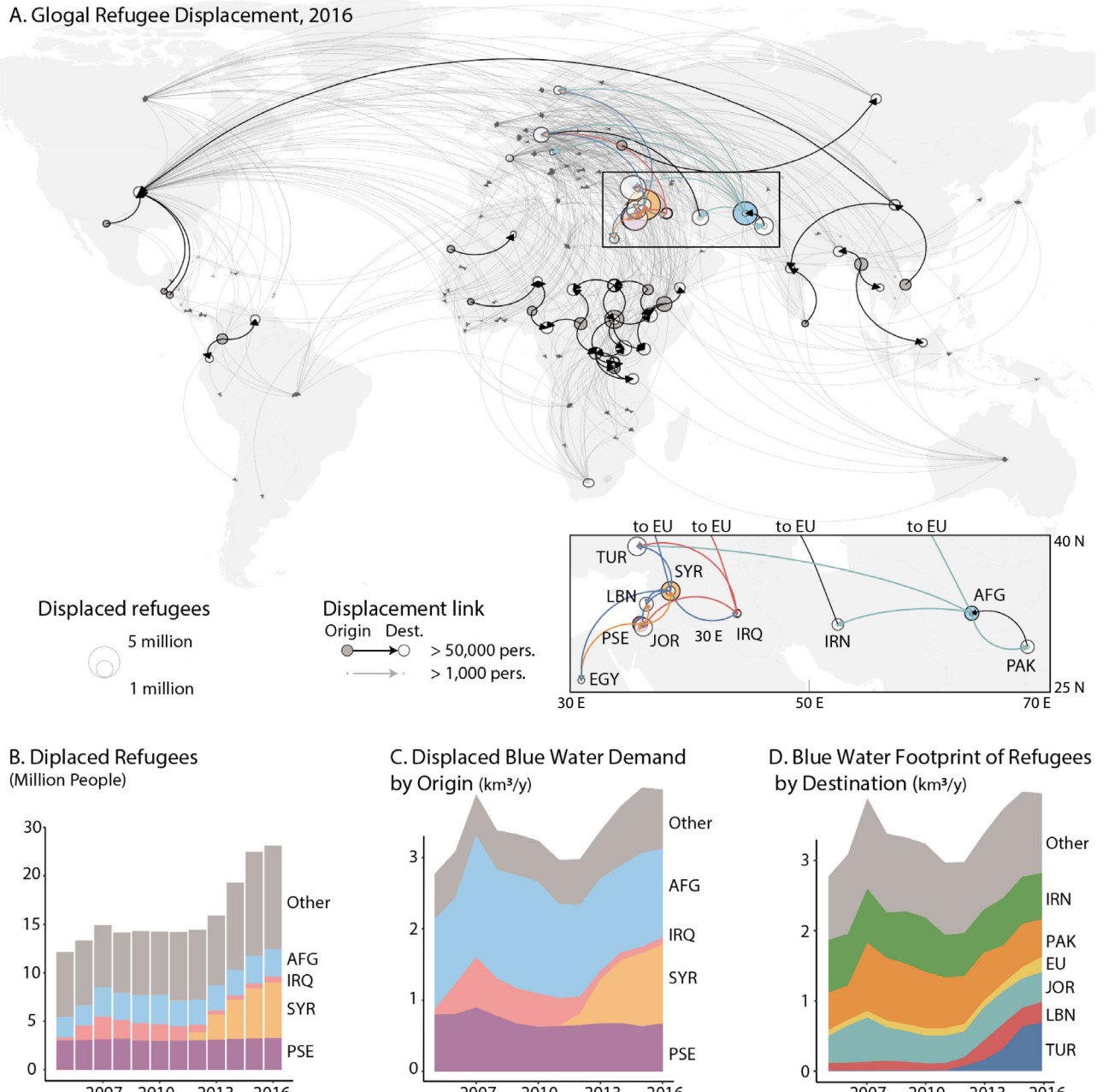

**Fig. 1 | Global refugee flows and displaced food and water demand. A** Origin and destination map of displaced refugees in 2016. Arrows indicate displaced populations of at least 1000 (gray) and 100,000 (black) people. Source for background map: R package 'maps' (v. 3.4.1). **B** Time series evolution of displaced refugees by country of origin. **C** Displaced Blue Water Demand associated with decreased food consumption in countries of origin. **D** Blue Water Footprint associated with the long-run food demand of refugees in host countries.

infrastructure[13,14] and the disruption of ecosystem services that support water provisioning, water distribution systems, flood management systems, and safe drinking water[15]. These disruptions of water security can have dramatic socioeconomic consequences, for instance, by affecting water prices and exacerbating preexisting economic inequalities and social divisions[2].

Yet the lion's share of a person's water consumption is embedded in the production of their food[16]. This notion is captured by the concept of per capita water footprint, which quantifies the volume of water necessary to produce, process, and distribute a person's annual food consumption[17]. Aggregated globally, this concept tracks the annual volume of water necessary to sustain humanity against the environmental limit of freshwater availability within which it can safely operate[18]. Water footprints can similarly be used to track countries'

progress towards the Sustainable Development Goal dealing with water stress (indicator 6.4.2)[19], although two important considerations are in order. First, per capita, water footprints vary substantially across countries as determined by prevailing dietary habits and food supply systems[16]. Second, the water stress of a country is determined by water withdrawals within its territory (which we here refer to as *water footprint*). These withdrawals are not necessarily equivalent to the water embedded in the food consumed by the population of that country (which we refer to as *water demand*). Indeed, the global food trade network allows water to be withdrawn in one country to produce food that is consumed in another country[20]. These flows of 'virtual' water between the origin and destination countries of traded food affect the global distribution of water resources and the water stress of nations[21]. Similar virtual water flows can be associated with human movements

(rather than traded goods) and have received much less research attention. Economic migrants have been shown to produce a flux of virtual water from the origin to the destination country, as the former ramps up export-bound production to supply expatriate communities with homegrown goods[22]. The situation is different for refugees, however, because food production in their home country is often severely disrupted, causing water demand to increase in the destination countries of refugees. The amplitude of this effect and its local and global (through trade) repercussions on water stress remain to be characterized.

We leverage recent data on the water footprint of the production and international trade of 370 food products[23] to estimate the refugee-related increase in the water demand of destination countries. This effort contributes to two distinct bodies of literature. The first contribution relates to the relationship between water resources and armed conflicts. Most recent research focuses on water (or lack thereof) as a driver, rather than a consequence, of armed conflicts[24]. While an increasing body of work evaluates the impact of warfare on the quality and quantity of local waters[25], recent work in Syria showed that conflict-related changes in land use can affect regional water availability beyond the battlefield[9]. Here, we show that the hydrological impacts of localized conflicts can be truly global through the effect of food trade and human migration. Our second contribution is to extend the concept of virtual water from its original application to supply chains and the water footprint of displaced *goods* to migrations and the water demand of displaced *people*. By keeping track of the ultimate origin of the water embedded in traded goods, we elucidate the direct (migration) and indirect (trade) effects of refugee displacement on country-level water stress. The water footprint dataset that we present captures these effects by uniquely distinguishing the effects of dietary habits, globalized supply chains, and agricultural water use efficiency.

We find that the water footprint of refugee displacements ($-31 \text{ km}^3 \text{ y}^{-1}$ in 2016) is disproportionately carried by a small number of countries where implications on water stress can be substantial. These countries face water scarcity conditions that are comparable to the origin countries of refugees. They tend to have less water-intensive food provision systems but more water-intensive diets. They also tend to predominantly rely on local water resources for food production, meaning that the transfer of water demand associated with refugee displacement toward destination countries is currently not substantially relieved by global trade. Leveraging these results, we examine the potential for international food supply chains, and refugee resettlement plans to alleviate the unevenly distributed water burden of global refugee displacements.

## Results and discussion
### Global footprint
We estimate the long-run water footprint of refugee displacements at nearly $31 \text{ km}^3 \text{ y}^{-1}$ in 2016, a 75% increase since 2005 (Fig. S2 in SI). This globally aggregated estimate was obtained by multiplying the number of displaced refugees by the per capita water footprint that we estimated for each destination country and each year (see Methods). It is approximately an order of magnitude smaller than that of economic migrants ($\approx 400 \text{ km}^3 \text{ y}^{-1}$) and about two orders of magnitude smaller than the volume of virtual water comprising global food trade ($\approx 2300 \text{ km}^3 \text{ y}^{-1}$). In per capita terms, the average water footprint of a refugee circa 2010 was approximately two-thirds that of an economic migrant, and the two estimates bracket the global average (Table 1). This discrepancy reflects an important difference, which is that economic migrants not only have a greater capacity for consumption but also tend to move to countries with greater consumption of water-intensive goods. In contrast, most refugee fluxes occur locally and connect countries with comparable (and lower than average) per capita water footprints.

**Table 1 | Per capita water footprint (WF) of migration in home and destination countries**

|  | Refugees | Economic migrants | Global population |
|---|---|---|---|
| Pop. (ca. 2010) | 10 M | 200 M | 6900 M |
| WF per cap., home | 1150 | 1572 | 1385 |
| WF per cap., dest. | 1438 | 2064 | – |

Population and average per capita water footprint (WF, in $[\text{m}^3 \text{person}^{-1} \text{y}^{-1}]$) for refugees (this study), economic migrants[22], and global population[77] in circa 2010. Water footprint estimates include blue and green water.

Water footprint estimates in Table 1 include both rainfed and irrigated agriculture. It excludes the so-called gray water that would be necessary to assimilate the agriculture-related pollutants released into the environment (not to be confused with wastewater reclaimed for irrigation, which is included in Table 1). Yet the water security implications of the displaced water demand are, to a large extent, determined by the destination country's reliance on so-called 'blue' water for irrigated agriculture. Blue water designates surface water, groundwater, and reclaimed wastewater that can be collected, stored, conveyed, and used as a production factor. Because the water used to meet the increased irrigation demand prevents it from being used by other potential end-users, the associated opportunity costs are high and conducive to water competition[26,27]. In contrast, rain-fed agriculture relies on 'green' water supplied by rain and stored as soil moisture before being used by crops. This water could not have been used for other productive purposes and has a little opportunity cost. We estimate the blue water footprint (BWF) associated with refugee displacement for each country as the blue water embedded in the food produced in this country but consumed by refugees anywhere. That is, it includes the (direct) blue water demand (BWD) of refugees arriving in that country and the indirect BWD transferred via international food trade to refugees displaced elsewhere (see Methods). The BWF will be negative if refugees move out of the country or if they move out of another country to which it exports food. Food also has an important cultural dimension, and newly displaced refugees tend to maintain strong ties to their native foods and traditional diets[28,29]. Because of this, recent refugees might have a consumption profile and per capita BWD that is different from the local population, although this difference will likely attenuate with time as migrants adopt some of the cultural habits of their new home[30]. The per capita BWD of refugees will also differ from that of their peers in their home country, who might have a similar consumption profile but are supplied by a different food provision system. We uniquely disentangle the effects of dietary habits and food provision systems (see Methods). We estimate the short-run BWD of refugees by combining the dietary habits of their origin country with the water intensity of the food provision system of their destination country. In contrast, in the long run, the BWD of refugees is determined by both the food provision system and the dietary habits of their destination country. In line with the Sustainable Development Goals Indicator 6.4.2, we then estimate the effect of refugee migration on water stress in destination countries by taking the ratio between (i) the BWF of global refugees in that country, which arises from the BWD of refugees in that country and its trade partners, and (ii) the country's available water resources, that is the difference between the country's total renewable water resources and its environmental flow requirements[19].

Overall, we find that refugees do not significantly contribute to overall water stress in most destination countries. With a few important exceptions that will be discussed, the blue water footprint of refugees accounts for less than 1% of total renewable water resources (net of environmental flow requirements) for most countries (see Table S2 in SI). There are, however, important differences between the food and water sectors of the origin and destination of refugee fluxes. We find that refugee displacement tends to transfer blue water

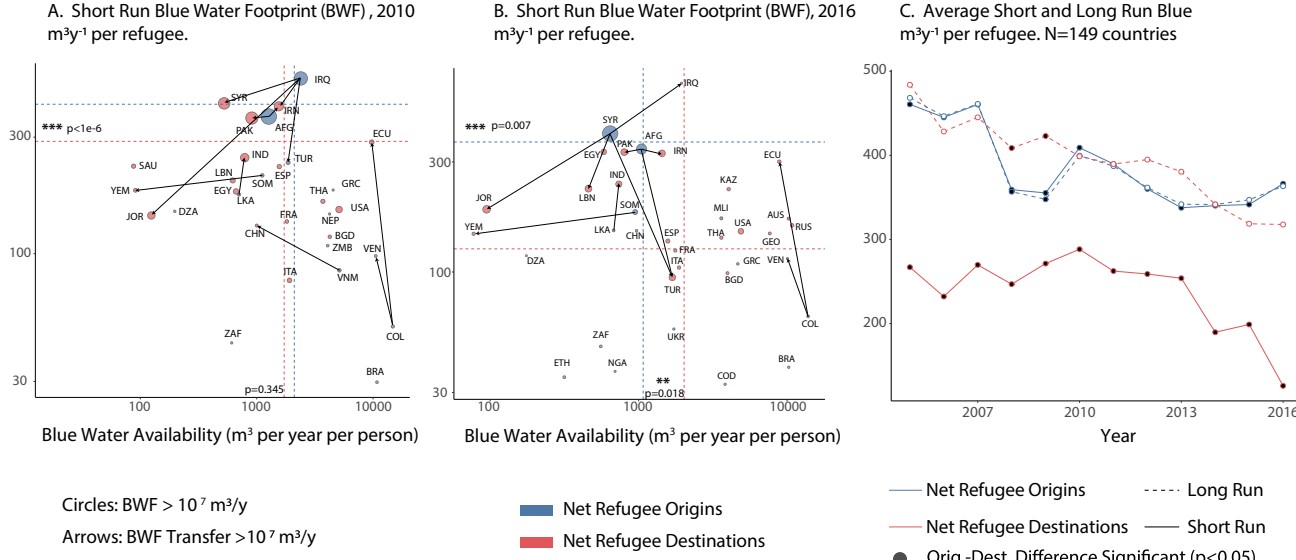

**Fig. 2 | Blue water footprint of refugees in origin and destination countries.**
**A**, **B** Average per capita blue water availability (*x*-axis) and short-run blue water demand (y-axis) of refugees in their countries of origin (blue) and destination (red) in 2010 (**A**) and 2016 (**B**). Symbol sizes are proportional to the net BWF of displaced refugees for the *N* = 27 (2010) and *N* = 30 (2016) countries with a net BWF larger than 0.01 km³/y in absolute value. Arrows indicate BWF transfers larger than 0.01 km³/y. Dashed lines indicate average axes values for origin (blue) and destination (red) countries, weighted by total net BWFs (symbol sizes), with asterisks indicating statistically significant differences (*p*-values:

$0.1 ^{***} < 0.01 < ^{**} < 0.05 < ^{*} < 0.1$, obtained through bootstrapped t-tests with 1000 repetitions). **C** Average per capita blue water demand per year across net origin (blue) and net destination (red) countries. Symbols represent weighted averages with net total BWF as weights. Black symbols represent significant differences (*p* < 0.05) between origin and destination countries. Dashed and solid lines in panel **C** respectively represent short and long-run results. Averages and t-tests in all panels were determined based on the full sample of *N* = 149 countries of our dataset. They include the BWFs that arise from the BWD of refugees in the country itself and in its trade partners.

demand toward countries with comparable levels of water availability but less water-intensive food provision systems. Indeed, we find no significant difference between average per capita water availability in origin and destinations for most of the study period (Fig. 2A, *x*-axis). The significant difference found in 2016 only appears twice in the 11 years period (Figure S3A in SI). Differences in food water intensity in origin and destination countries can be seen by considering differences in short-run BWD (Fig. 2A, B, *y*-axis), which keep dietary habits constant. Refugees tend to move to destinations with significantly less water-intensive food provision systems throughout the study period (Fig. 2C, dashed). This result is driven by refugees fleeing Iraq (before 2011, Fig. 2A), Afghanistan, and Syria (after 2011, Fig. 2B). These three countries are the home countries of a sizeable portion of refugees and have some of the world's largest per capita BWD. However, the consumption profiles in destination countries have comparatively more water-intensive foods, so the short-term gains in water efficiency from refugees moving to countries with less water-intensive food provision systems are likely more than compensated by dietary changes in the long run (Fig. 2C, solid lines). Whether the blue water demand of incoming refugees translates into additional blue water withdrawals depends on the extent to which the destination country relies on in-kind food assistance or engages in the global food trade. Such international transfers of virtual water can either alleviate water stress by distributing it from water-vulnerable countries outwards towards the global market[26,31,32] or, alternatively, propagate water demand to exacerbate tensions in already water-stressed export countries[33,34]. We find that countries that produce or host the most refugees tend to rely on domestically sourced blue water for the large majority (>80%, Fig. S3B in SI) of their food water needs, and regression results presented in Table S6 (in SI) show that reliance on virtual water imports does not increase significantly with the number of refugees in destination countries. Together, these results suggest that international trade—currently—plays a limited role in alleviating or propagating any water stress associated with refugees. Note that we were not able to

include in-kind food assistance in the analysis, but discuss evidence in the Methods section that in-kind international food assistance is unlikely to have a significant effect on the per capita BWD of refugees.

## Regional impacts

The globally averaged results discussed above mask substantial variations between countries. One-half to two-thirds of the blue water demand displaced by refugees in any given year during the 2005–2016 period can be traced back to conflicts in Afghanistan and Syria (Fig. 1C) and the associated displacement of at least 8.8 million refugees from these two countries. More than 95% of the blue water demand associated with these two conflicts was transferred to six destination regions (Pakistan, Iran, Turkey, Lebanon, Jordan, and the European Union, Fig. 1D) with three distinct types of water stress implications.

The first group of countries has a large enough agriculture sector to accommodate the food and water demand of refugees with no significant impact on country-level water stress. The majority of refugees displaced at the height of each crisis were hosted by Pakistan (1.9 million Afghans in 2010), Iran (1.0 million Afghans in 2010), and Turkey (2.8 Syrians in 2016). All three countries import very little food and rely heavily on domestically extracted blue water (Fig. SI S4B), with per capita blue water demands in the top 12% of the 173 countries in our dataset (see SI Fig. S4A). The blue water demand transferred by refugee displacement to each country is substantial and reaches 0.65 to 0.8 km³ per year in the long run (Table 2). For comparison, this is approximately equivalent to three-quarters of the total annual volume discharged by the Jordan River under natural conditions[35]. Yet, in relative terms, the blue water demand displaced by refugees represents less than 1% of each country's available water resources and has a negligible effect on country-level water stress (Table 2). It is important to remember that these country-level outcomes overlook potentially large variations in the distribution of refugees within the destination countries. More than 80% of Afghan refugees in Pakistan settled in arid

**Table 2 | Contribution of refugee migration to water stress in key destination regions**

| Dest. | Pop. | Stress | Refug. | Δ BWD | | Δ Stress | |
|---|---|---|---|---|---|---|---|
| | 10⁶ | % | 10⁶ | MCM/y | | %-points | |
| | | | | SR | LR | SR | LR |
| IRN | 73.8 | 81 | 1.0 | 406 | 740 | 0.6 | 1.0 |
| PAK | 179.4 | 115 | 1.9 | 683 | 806 | 0.6 | 0.7 |
| TUR | 79.8 | 42 | 2.8 | 282 | 651 | 0.3 | 0.7 |
| EU | 485.0 | 21 | 0.8 | 79 | 83 | 0.0 | 0.0 |
| LBN | 6.7 | 59 | 1.7 | 390 | 330 | 12.4 | 7.9 |
| JOR | 9.6 | 100 | 3.3 | 631 | 495 | 75.2 | 47.4 |
| JOR* | 9.6 | 100 | 1.1 | 217 | 162 | 24.2 | 13.5 |

Population, water stress, and refugees in Pakistan, Iran, Turkey, the European Union, Lebanon, and Jordan.
The estimated short-run (SR) and long-run (LR) water footprint of refugees (ΔBWF, in a million cubic meters per year, note that 1 km³ = 1000 MCM) are given in each destination country, along with its effect on water stress (Δ Stress). PAK and IRN focus on 2010, at the height of the Afghan refugee displacement, and only include Afghan refugees—results including all refugee origins are provided in Table S2. Data for the EU are obtained for 2016 and include Afghan and Syrian refugees. LBN and JOR include estimates of Syrian refugees (registered and unregistered) obtained from refs. 10, 43 for Lebanon and Jordan, respectively. LBN and JOR also include registered refugees from all the other countries in the UNHCR dataset for 2016, which includes Palestinians under the UNRWA mandate. JOR* only includes registered and unregistered Syrian refugees from ref. 10 and estimates the net effect of the Syrian refugee crisis on Jordanian water stress by subtracting the exploitable increase in transboundary river flow (18 MCM/y) from ΔBWF when determining ΔStress.

Khyber Pakhtunkhwa and Balochistan provinces, which together account for less than 20% of the country's population and cropland (see SI Fig. S6). The increased food demand is likely supplied by the (relatively) more blue-water-abundant Indus Valley, where the majority of Pakistan's irrigated cropland is located. Yet, unlike the virtual water embedded in food, the physical water used for the domestic needs of refugees (e.g., drinking and bathing) cannot be imported from more water-abundant parts of the country and has to be extracted locally, which can impose a significant strain on local infrastructure and water resources[36]. These challenges arise within broader water issues in Iran and Pakistan, which are both major exporters of blue water through global food trade amidst rapidly depleting groundwater resources[37,38]. Both countries are facing major water scarcity challenges[39,40] that are little affected by the food and water demand of refugees.

Refugees have a similarly negligible impact on country-level water stress in the second group of nations, this time due to their comparatively lower reliance on blue water for food. European food systems rely heavily on rain-fed agriculture, with blue water only accounting for approximately 6% of per capita food water footprints. The 1.1 million refugees from Syria and Afghanistan in the European Union (EU) countries by 2016 have only displaced about 0.083 km³/yr of blue water demand toward the EU in the long run (Table 2). This is approximately 10 times smaller than that in Iran, Pakistan, or Turkey for comparable fluxes of incoming refugees. In contrast to Iran and Pakistan, a substantial portion of European food is imported and does not deplete blue water resources within the continent. Therefore, although the political, cultural, and socioeconomic implications of refugees in Europe are well documented[41,42], water stress associated with displaced food demand is unlikely to be a major issue.

In the third group of countries, however, migration causes a demand displacement that is sizable relative to water availability, and water stress implications can be dramatic. While specific numbers vary across sources (see Methods and Table 2), approximately 1.2 and 1.1 million Syrians had respectively crossed into Lebanon and Jordan (respectively) by 2016[10,43]. With a substantial population of refugees even before the Syrian crisis, both countries have among the largest per capita concentrations of refugees in the world amidst high to very high water stress conditions (Fig. S5). In Lebanon, a country that is historically water-abundant compared to the regional average, water stress increased by 24 percentage points (from 35% to 59%) between 2005 and 2015[44]. We estimate that approximately half (12.4 percentage points, Table 2, LBN) of this increase arises from the food-related short-run BWF of displaced refugees. In Jordan, freshwater withdrawals have long exceeded renewable water resources[45], and the country is facing a severe unfolding water crisis that is exacerbated by climate change[10,11]. The multi-faceted challenges faced by the Jordanian water sector are well documented[10,45–50] and include (among many other issues) the need to supply water to an increasing population of refugees[51]. Unlike previous efforts, this study documents the *food* demand displaced by refugee migrations into Jordan, which we estimate has contributed 47–75 percentage points to the country's overall water stress (Table 2, JOR short and long run). For comparison, countries with water stress *levels* beyond 40% are generally considered subject to high water stress[52].

Focusing on the subset of refugees in Jordan that are from Syria and registered with UNHCR allows us to compare the demand- and supply-side effects of refugee displacement on water resources. On the demand side, we associate registered Syrian refugees in Jordan with a short-run increase in blue water demand of 217 million cubic meters (MCM, 1 km³ = 1000 MCM) in 2016 and 109 MCM in 2015. Note that this is a conservative estimate because a substantial fraction of Syrian refugees in Jordan is unregistered. Approximately 40% of that increased demand was covered by food imports, mostly from Egypt, Saudi Arabia, and Syria (Fig. 3A), leaving a blue water demand of at least 65 MCM to be covered by Jordanian water resources in 2015 (Fig. 3A, teal area). On the supply side, average Yarmouk river flow from Syria into Jordan has increased by about 50 MCM/yr between 2011 and 2015 (compared to pre-2011 levels, Fig. 3A, dashed), a sudden increase that occurred on the heels of decades of steadily decreasing flows (Figure S7) and a history of water allocation disputes between Jordan and Syria[46]. A recent estimate suggests that approximately 55% of the observed flow increase can be attributed to abandoned irrigation agriculture in upstream Syria by fleeing refugees[9,47] (Fig. 3A, dotted). Based on a comprehensive model of the Jordanian water sector[10], we estimate that approximately 80% (or 18 MCM) of the Yarmouk increase attributed to refugees was used by Jordanian agriculture in 2015 (see SI). This increase in blue water supply only offsets about 39% of the blue water demand associated with registered Syrian refugees in Jordan (Fig. 3A, solid line vs. teal area). Overall, we estimate that refugee migration from the war in Syria has alone increased water stress in Jordan by approximately 13.5–24.2 percentage points (Table 2, JOR*).

It is important to keep in mind that the above discussion focuses on the blue water embedded in food and excludes the additional effect of domestic (drinking, cleaning, etc.) water consumption. A recent estimate in Lebanon[43] associates Syrian and Palestinian refugees with a 20% increase in domestic water consumption and a 3 percentage point increase in country-level water stress. This increase adds to the 12.4 percentage point increase in water stress that we found for the food and water demand of refugees in Lebanon in the short run (Table 2, LBN). In Jordan, increased domestic water demand associated with Syrian refugees will, alone, cause an estimated 5.7 percentage point increase in the fraction of water-vulnerable households through the end of the century[10]. Furthermore, the reduction of blue water security is likely a slow process eroding water resilience with potentially dramatic and unforeseen consequences in the future. To be sure, the water challenges in Jordan and Lebanon predate the arrival of refugees who highlighted, rather than created, long-standing issues in both countries' water sectors. However, the water security implications of refugees have shaped the national discourse around water governance in both countries[13,53] and spurred social unrest. As Baylouny and Klingseis[54] put it: "Syrian refugees, in effect, have been catalysts of

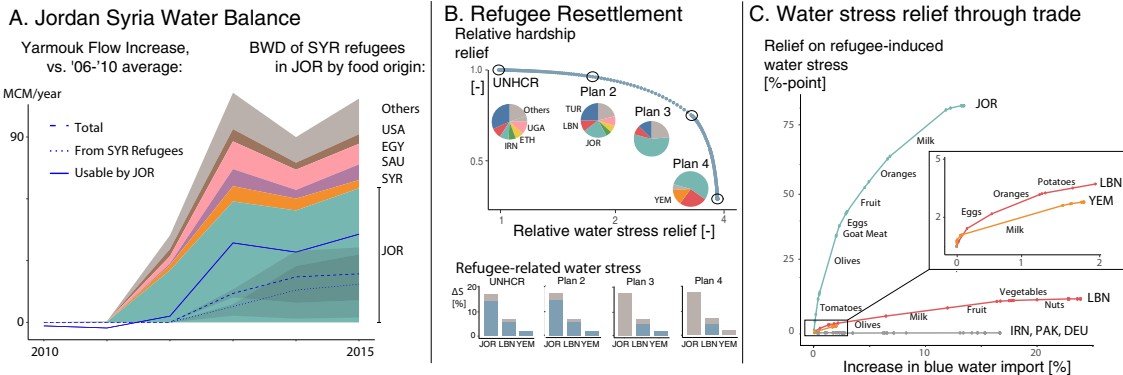

**Fig. 3 | Supply-side effect, refugee resettlement, and food import. A** Changes in the annual flow volume of the Yarmouk River compared to the 2006–2010 average. Blue lines indicate the observed flow changes at the Al Wehda dam on the Syria–Jordan border (dashed), the estimated portion of that change attributable to abandoned Syrian agriculture (dotted), and the portion of the refugee-attributed flow increase that was likely retrieved for Jordanian irrigation (plain), with shaded area indicating approximate confident ranges (see SI). Stacked colors indicate the blue water demand of Syrian refugees displaced into Jordan. Approximately 60% of this water is not procured through food import but sourced domestically by Jordan (teal). **B** Refugee resettlement trade-off between relieving the individual hardship of refugees and relieving the water stress of the countries of refuge. Dots in the main graph represent alternative resettlement plans, where the 1.44 million refugees in the current United Nations High Commissioner for Refugees (UNHCR) plan are selected from alternative combinations of current host countries. Pie charts represent the compositions of the highlighted plans. Bar charts represent water stress associated with UNHCR-registered refugees in Jordan, Lebanon, and Yemen without resettlement (gray) and with each highlighted plan (blue). **C** Relative increase in blue water import (y-axis) necessary to relieve the water stress (x-axis) associated with the short-run food consumption of refugees. The relationship is displayed by the type of good included in the FAO balance sheet in decreasing order of blue water intensity. Note that the analyses in Panels **B** and **C** only include registered refugees under the UNCHR mandate, which are eligible to be considered for resettlement.

domestic conflict over water security, providing one spillover from Syria's civil war".

## Refugee resettlement and food import

Our analysis has shown that the water stress of refugee displacements is disproportionately borne by a few countries that host an incommensurate number of refugees amidst severe preexisting water security challenges. This burden exacerbates a broader landscape of water inequality and injustice in the Global South, where resource mismanagement[5], poor governance[55,56], the commodification of water and land[57], extractive industries[58], environmental cost shifting[33], and the legacies of colonialism[59] often amplify water scarcity in arid and semi-arid regions that will be strongly impacted by climate change[60]. We recognize that mitigating these issues is a complex undertaking that requires an appreciation for the 'relational, situated, and context-sensitive rather than universalistic' nature of water justice[61,62] and that focusing on the water stress of refugee displacements is inevitably a partial—and somewhat reductionist—perspective. With these caveats in mind, we investigate the potential to alleviate this increased water burden, either by transferring virtual water demand *away* from water-vulnerable host countries using refugee resettlement programs; or by transferring virtual water supply *toward* water-vulnerable countries through enhanced food imports.

Refugee resettlement is actively promoted by UNHCR as a mechanism to both protect vulnerable refugees and provide a long-term durable accommodation[63]. As an added benefit, this process might help alleviate water stress in current host countries, as refugees are expected to resettle in generally water-abundant destination countries of the Global North. As of 2021, the UNHCR has identified approximately 1.44 million refugees in need of resettlement based on current individual hardship[64], predominantly in Turkey, Uganda, Lebanon, and Ethiopia (Fig. 3B UNHCR plan, pie). Using blue water demand estimates for 2016, we estimate that resettling these refugees to water-abundant countries in Europe or North America would alleviate approximately 99.7 MCM/y of blue water demand in current countries of refuge, which only corresponds to a decrease in water stress of at most 2 percentage points in current countries of refuge (Fig. 3B UNHCR plan, bars). A larger relief on water stress could be obtained by relocating a larger share of refugees out of the most water-stressed host countries, but this would come at the expense of dampening the overall benefit of resettlement in terms of relieving the individual hardship faced by the refugees. Indeed, UNHCR currently identifies refugees for resettlement based on their individual circumstances, meaning that the current plan theoretically allows for the largest possible decrease in the average hardship faced by individual refugees. We characterize this trade-off by identifying the set of resettlement plans that are Pareto optimal (Fig. 3B) in the sense that no other plan could simultaneously relieve more water stress and more individual hardship. Our approach hinges on the fact that the (to us) unobserved selection criteria used by UNHCR in their current resettlement plan is based on the hardship faced by individual refugees (which we express in terms of waiting time, see Methods) rather than the burden carried by the current countries of refuge[63]. We found that relatively small changes in current resettlement plans can have a potentially outsize benefit in terms of relieving the water stress of refugee countries. For example, relocating about 15% of the 1.44M refugees of the UNHCR plan out of Jordan and Lebanon instead of Turkey, Uganda, Ethiopia, and Iran would nearly double the average relief of water stress while decreasing the average relief of individual hardship by about 5% (Fig. 3B, compare UNHCR plan to plan 2). Despite this potential, weighing the individual hardships faced by refugees against the collective burden shouldered by the refuge countries is a thorny endeavor. Currently, UNHCR explicitly states that 'resettlement should not be pursued because individual refugees have become a burden'([63], p. 245). However, the hardship faced by refugees is likely determined, to a certain extent, by the ability of the country of refuge to accommodate them. Whether and to what extent this ability should be accounted for in refugee resettlement processes is, at the end of the day, a political and ethical decision that can be informed by the Pareto optimization framework that we describe.

While indicative of the potential water benefit of refugee resettlement, the above analysis should be seen in its proper context. The long-distance resettlement of refugees, whose lives are already upended by war and displacement, is a complex issue affecting the lives of millions of real persons. Displacing goods rather than people is

perhaps comparatively less challenging. Instead of resettling refugees to more water-abundant host countries, water-abundant countries can export a greater share of water-intensive food to current countries of refuge and thereby relieve water stress by lessening demands on local food production and water use[65]. Completely alleviating the short-run water stress of refugees in Jordan and Lebanon would respectively require a 14% and 24% increase in blue water imports from current trade partners (Fig. 3C). However, there are substantial variations between goods, for which the relation between increased blue water imports and the ensuing relief in water stress is strongly non-linear (Fig. 3C). In Jordan, a mere 2% increase in current blue water imports that would specifically focus on wheat, poultry, tomatoes, and olives can alleviate nearly 30 percentage points of water stress. This represents a larger relief than what could be obtained by relocating *all* 0.7 million refugees in Jordan that are registered with UNHCR and thus potentially eligible for resettlement. Overall, our results suggest that enhanced imports of water-intensive goods—either through food aid or international trade—could disproportionately relieve the short-run effect of refugee food demand on water stress. This is particularly true for Jordan and Lebanon, where a substantial (respectively 64% and 75%) portion of the blue water embedded in the food consumed by refugees is currently sourced domestically despite severe water scarcity.

## Policy implications

Global refugee displacement is a complex multi-dimensional crisis that is upending the lives of millions of real humans. Of course, focusing on water-related impacts (even when accounting for 'refugee hardship', as in Fig. 3B) cannot fully capture the myriad of challenges faced by both the refugees themselves and by their countries of refuge. To be clear, we are not suggesting that water stress should be singled out as a basis for trade, aid, and migration policy, which is driven by a range of social, economic, cultural, political, and ethical factors that have nothing to do with water[66]. Similarly, the objective of water management should not necessarily be to minimize the use of water resources but rather to ensure that the value of water for society and the environment is maximized[67]. Rather, our analysis positions itself within the framework of the Sustainable Development Goals, where water stress reduction (SDG indicator 6.4.2) is one of many indicators serving to monitor progress towards a broader set of sectoral goals in a way that embraces (or at least acknowledges) their complex interactions. More broadly, we believe that our results are policy-relevant in terms of (i) identifying emerging water security challenges in key regions of concern and (ii) characterizing the potential for existing mechanisms—specifically, refugee resettlement programs and international food supply chains— to alleviate them. We also believe that the quantitative estimate of virtual water fluxes associated with forced population displacement that we provide is helpful in informing more complex operational models to support policy (see, e.g.,[10] for Jordan).

Reassuringly, our results have shown that water stress associated with increased food consumption is not a prominent issue in the overwhelming majority of destination countries. This finding is important in the context of increasing rhetoric from the far-right that presents migrants and refugees as draining the resources of host countries (see, e.g.,[68] for the European Union). In a few specific countries, however, the added water demand associated with the food consumed by refugees has the potential to destabilize an already overextended water sector. Along with interventions to improve the water use efficiency of local food production (e.g.,[69]), enhanced import of water-intensive foods from water-abundant countries, either through food aid or favorable trade conditions, stand out as promising approaches to mitigate impending water crises in the short run. Yet international food supply chains are themselves vulnerable to conflict-related shocks, as illustrated by the ongoing crisis in Ukraine. A resilient trade network, where countries can respond to shocks by forming new partnerships, is essential to avert the cascading effects of conflicts on food security. Such networks are facilitated by policies that support the diversification of import sources[70]. In the long run, the safe repatriation of refugees in their country of origin—or their long-term relocation to countries where they can sustainably build a new life— stand out as important aspirational goals. Refugee resettlement can simultaneously provide a lasting solution for the refugees themselves while relieving the often resource-poor (both in terms of water and economic resources) countries that currently shelter them. Our analysis illustrates the dynamics of this win-win proposition in the specific context of water stress. Yet, as of 2021, less than 3% of the 1.44 million refugees of the UNHCR plan have been successfully relocated[64]. There is a salient need—and a moral imperative—for non-water stressed higher-income countries to support the resettlement effort through increased resettlement pledges.

## Methods
### Notation

We use upper and lower case variables to represent absolute and per capita quantities, respectively. Variables in prime notation ($\cdot'$) are used to represent quantities of food (tonnes) to distinguish them from volumes of virtual water ($m^3$). Subscript are used to indicate the type of food ($k$), the country of production of the food ($p$) and the countries of origin ($o$) and destination ($d$) of the refugees. Superscript $y$ indicates the considered year in the 2005-2016 period. Throughout the paper, we also refer to blue water demand (BWD) of refugees as the volume of water embedded in the food consumed by refugees in their country of destination, no matter where this water was extracted. In contrast, we refer to blue water footprints (BWF) of refugees as the volume of water extracted from a given country to produce the food that they consume, no matter where these refugees are hosted.

### Refugees

Refugee displacement matrices $R_{od}^{y}$ were constructed using UNHCR data (freely available at https://www.unhcr.org/refugee-statistics/) and represent the number of refugees and asylum seekers from country $o$ living in country $d$ on year $y$. The dataset includes international refugees that seek refuge outside of their country of origin and are included under UNHCR or UNRWA mandates. Unless otherwise noted, we exclude migrants that are not registered with a UN agency. Approximately 1.1 and 1.2 million Syrian refugees were reported respectively in Jordan[10] and Lebanon[43] in 2016, of whom 0.65 and 1.0 million (respectively) are registered and included in UNHCR data. For these two countries in particular, we included unregistered refugees in country-specific results when indicated (i.e., Table 2 and Fig. 3).

We focus on international refugees for consistency with the virtual water data set that provides water footprint estimates at the country level. As discussed in the main text, this scale of analysis relies on the premise of a working domestic food supply system that conveys food from the agricultural production regions of the country to the marginal lands where refugees are often hosted[71]. This assumption generally does not hold for internally displaced persons (IDP) that are displaced within their own war-torn country. IDPs outnumber international refugees by nearly two to one[12] but had to be excluded from our analysis due to unavailable water footprint data at the sub-national scale. Recent research using satellite imagery finds evidence of enhanced land use change in the vicinity of IDP camps[72,73], suggesting a sizable local impact on food and water availability that remains to be characterized.

We focus on the 2005–2016 period, which overlaps with the virtual water data set. We removed countries hosting (or supplying) less than 1000 refugees in all 11 years of the 2005–2016 study period. The remaining set of 167 countries and autonomous regions that contributed meaningfully to global refugee migration during that period were included in the

analysis. Data limitations prevented us from determining food water demand or water stress conditions for 39 of these countries (see Table S1). The water demand transfers associated with refugees originating from these countries were estimated in an identical way to refugees that are stateless or of unknown origin. Namely, the short-range per capita water demand of refugees from these countries in their countries of destination was approximated as the weighted average of the corresponding values of all other refugees migrating into the same country. The long-range per capita water demand is solely determined by the country of destination and not affected by the missing water data in the country of origin. However, we were not able to determine the water demand transfer associated with refugees migrating *into* the 39 countries with missing water data. Approximately 5% of global refugees migrating into these countries (Table S1) were therefore excluded from the analysis.

### Long-run food–water demand

The water footprint of primary and processed crops was obtained from the CWASI dataset[23] (freely available at https://www.watertofood.org/download/). The dataset combines trade data from the Food and Agriculture Organization (FAO) with a model estimating the crop-specific water requirement on production sites in the countries where the food was produced[74,75].

Unlike previous data, water footprint estimates are provided *yearly* between 1961 and 2016, based on the assumption the variability of virtual water fluxes is driven by variations in crop yields and traded volumes rather than climate (evapotranspiration) variability[76], although the latter is indirectly included through crop yields. As such, it accounts for technological improvements, for instance, pertaining to crop varieties, fertilizers, or irrigation techniques. The dataset provides the total 'virtual' water embedded in the production and international trade of 370 raw and processed foods. It also quantifies the 'blue' water footprint as the irrigation demand required to compensate for rainfall shortages, notwithstanding the source of irrigation water, which can be surface water, groundwater, or wastewater. The dataset excludes the so-called gray water footprint, which refers to the water needed to dissolve the pollutants associated with food production. The gray water footprint makes up a non-negligible portion (9%[77]) of the water footprint of global food production but is challenging to systematically estimate accurately at the global scale[78,79].

We processed the data as detailed in SI to obtain $c_{dk}^{(y)}$, the per-capita (blue or total) water footprint of food type $k$ consumed (though not necessarily produced) in country $d$ on year $y$. We also computed $X_{pdk}^{(y)}$, the fraction of the virtual water (blue or total) in food type $k$ consumed in country $d$ that ultimately originates from country $p$ through international food trade (see SI). The water footprint (blue or total) in country $p$ of a (native) person consuming food in country $d$ in year $y$ is then:

$$\omega_{odp,\mathrm{LR}}^{(y)} = \sum_k c_{dk}^{(y)} X_{pdk}^{(y)} \tag{1}$$

Note that the index $p$ represents the country where the water embedded in the consumed food was extracted. Because of international food trade, this country can differ from the country of refuge $d$ where the food is consumed. Equation 1 represents the *long-run* per capita water footprint in country $p$ of a refugee from *any* country $o$ that migrates into country $d$. It assumes that the consumption habits of refugees are indistinguishable from that of the native population in the country of refuge after a sufficiently long period of time.

### Short-run food–water demand

This assumption might not hold over shorter time horizons, where the per capita water demand of refugees might differ substantially from that of the native population. Assuming that newly arrived refugees will preserve dietary habits from their country of origin $o$ but consume food obtained in their country of refuge $d$, the short-run water footprint in country $p$ of a refugee from country $o$ that lives in country $d$ can be expressed as:

$$\omega_{odp,\mathrm{SR}}^{(y)} = \sum_k c_{odk}^{(y)} X_{pdk}^{(y)} = \sum_k c_{ok}^{'(y)} W_{dk}^{(y)} X_{pdk}^{(y)} \tag{2}$$

Note that 'long-run' and 'short-run' are here (loosely) used to express situations where migrants adopt the dietary habits of their country of refuge or, respectively, preserve the dietary habits of their country of origin. The time taken to transition from the latter to the former might vary substantially across contexts. The above Equation, $c_{ok}^{'(y)}$ denotes the mass (*tonne*) of good $k$ that a refugee would have consumed in their country of origin $o$. To estimate that value, we used country-level food production, trade, and stock variations from the Food and Agricultural Organization (FAO) food balance sheets as described in SI. Equation 2, $W_{dk}^{(y)}$ denotes the virtual water content ($m^3$ of virtual water per *ton*) of food $k$ obtained in country $d$, which we estimate from the CWASI dataset as described in SI.

Because we were not able to obtain globally consistent data on in-kind food assistance that covers the 2005-2016 period, we assume that the water intensity $W_{dk}^{(y)}$ of the food consumed by the refugees is identical to that of the local population in their country of refuge. Because international food assistance attenuates refugees' reliance on locally produced food, neglecting it might cause us to overestimate the short-run impact of refugees on the water stress of destination countries. However, we do not believe that this overestimation is substantial for two reasons. First, one-half[80] to three quarters[81] of refugees do not live in refugee camps, and so are less likely to be reached by in-kind food assistance, which has rapidly been phased out over the last 20 years and replaced by cash assistance[82]. Second, a non-negligible portion of food rations is sold into the black market. Ethnographic data from Kakuma camp in Kenya[83] suggest that 80% of refugees with access to cash (through employment or remittance) sold nearly all their relief packages into the black market. The remaining 20% who depend almost wholly on relief packages still sold nearly 50% of their allotment to traders. These findings suggest that even refugees benefiting from in-kind food assistance are likely to procure a substantial part of their food from the same sources as native inhabitants. In other words, $W_{dk}^{(y)}$ is unlikely to be a substantially different between refugees and the local population. Even if $W_{dk}^{(y)}$ were identical for refugees and locals, in-kind food aid is excluded from the international trade data used in this analysis. If the virtual water imported through in-kind food aid is substantial compared to the country's water footprint of food consumption, neglecting it can introduce a nonnegligible error in $W_{dk}^{(y)}$. However, the virtual water (blue, green, and gray) imported through in-kind food aid reported in ref. 84 remains below 9% of the water footprint of the national consumption of crop products[85] for the 9 receiving countries that account for 2% or more of the global water footprint of food aid (Table S3). This suggests that the effect of in-kind food aid on the average virtual content of food in destination countries is not substantial.

Our assumption that recently arrived refugees preserve their dietary habits from their country of origin is broadly supported by ethnographic evidence. For example, research in refugee camps in Kenya[86] and Rwanda[80] suggests that nearly all refugees sell part or most of their food allotment on the black market to buy food that conforms with their traditional dietary habits (thus creating a sense of normalcy[86]). However, it is important to note that the assumption neglects any potential coping mechanisms to address prevailing conditions of food insecurity[87]. Neglecting these coping mechanisms (e.g., eating less or less diverse[88]) might cause us to overestimate the BWF of displaced refugees in food-insecure conditions.

## Displaced blue water demand and water stress

The blue water footprint in country $p$ of the food consumed by refugees displaced from country $o$ into country $d$ in year $y$ is expressed as:

$$\Delta \text{BWF}^{(y)}_{odp,\text{SR or LR}} = R^y_{od} \cdot \omega^{(y)}_{odp,\text{SR or LR}} \qquad (3)$$

If $p = d$, this represents the increased demand for the local water resources of the country of refuge $d$. If $p \neq d$, Equation 3 represents the water embedded in increased food export from a third country $p$ associated with increased food demand in the country of refuge $d$. Note that $\Delta \text{BWF}^{(y)}_{odp,\text{SR or LR}}$ can be negative if refugee displacement decreases water demand in country $p$, either because refugees migrate out of $p$ (i.e., $p = o$) or because $p$ is a major exporter of food to $o$. Accordingly, the aggregate blue water demand transferred by refugees moving out of $o$ (to any destination, Fig. 1C) or into $d$ (from any origin, Fig. 1D) are respectively expressed as:

$$\Delta \text{BWD}^{(y)}_{o,\text{SR or LR}} = \sum_d \sum_p R^y_{od} \cdot \omega^{(y)}_{odp,\text{SR or LR}} \qquad (4)$$

$$\Delta \text{BWD}^{(y)}_{d,\text{SR or LR}} = \sum_o \sum_p R^y_{od} \cdot \omega^{(y)}_{odp,\text{SR or LR}} \qquad (5)$$

Similarly, the blue water footprint in country $p$ of global refugee displacement (from any origin or destination, Fig. 2A) is expressed as:

$$\Delta \text{BWF}^{(y)}_{p,\text{SR or LR}} = \sum_o \sum_d R^y_{od} \cdot \omega^{(y)}_{odp,\text{SR or LR}} \qquad (6)$$

All above expressions can be similarly used for total water footprints (Fig. 1C and Table 1) by replacing the per capita blue water demands ($\omega_{odp,\text{SR or LR}}$) with the *sum* of blue and green water demands.

In line with the UN SDG indicator 6.4.2, we define blue water availability as the difference between total renewable water resources (TRWR) and environmental flow requirements (EFR), which we normalize by the country's population size to obtain per capita blue water availability displayed in Fig. 2A and B. SDG indicator 6.4.2 then defines water stress in country $p$ on year $y$ as the ratio between total freshwater withdrawals (TFWW) and blue water availability[89]:

$$S^{(y)}_p = \frac{\text{TFWW}^{(y)}_p}{\text{TRWR}^{(y)}_p - \text{EFR}^{(y)}_p} \qquad (7)$$

We extend this framework to evaluate the change in water stress due to the increased (or decreased) blue water footprint associated with global refugee displacements:

$$\Delta S^{(y)}_{p,\text{SR or LR}} = \frac{\Delta \text{BWF}^{(y)}_{p,\text{SR or LR}} \cdot \frac{1}{\epsilon^{(y)}_p}}{\text{TRWR}^{(y)}_p - \text{EFR}^{(y)}_p} \qquad (8)$$

where $\epsilon$ represents irrigation efficiency, which is necessary to relate the gross water use metric in the water stress indicator (TFWW) to the net water consumption metric ($\Delta \text{BWF}$) obtained in our analysis. Following ref. 90, we estimated $\epsilon$ as the ratio between a country's irrigation water requirements (IWR) and its irrigation water withdrawal (IWW). All above country-level metrics (population, TFWW, TRWR, EFR, IWR, and IWW) were obtained from the FAO AQUASTAT database (openly available at https://www.fao.org/aquastat/en/) and linearly interpolated to obtain annual estimates. We used agricultural water withdrawals for countries with no IWW estimates[90] and set EFR = 0 for the 10 (mostly arid) countries with no provided value (Table S2). Countries without data for any of the other metrics were excluded from the analysis (see Table S2).

## Refugee resettlement and water stress relief

Refugee resettlement decisions entail a trade-off between relieving the individual hardship faced by the refugee themselves and relieving water stress in the countries currently hosting them. To characterize this trade-off, we express the short-run effect of a marginal refugee on the water stress conditions in their country of refuge $d$ as:

$$\Delta s^{(y)}_{d,\text{SR}} = \frac{\sum_o \omega^{(y)}_{odd,\text{SR}} \cdot \frac{1}{\epsilon^{(y)}_d}}{\text{TRWR}^{(y)}_d - \text{EFR}^{(y)}_d} \qquad (9)$$

We compute $\Delta s^{(y)}_{d,\text{SR}}$ for 2016 (the last year of the CWASI dataset) and use it as a proxy for its value in 2020 (the year of the considered resettlement plan). The above equation assumes that all refugees moving into country $d$ (no matter their origin) have an identical marginal blue water footprint that is equal to their average per capita blue water footprint in their country of refuge. This assumption implies that the water 'released' by resettling refugees will fully serve to relieve water stress in their former country of refuge and not instead be used by other water users. In other words, we assume that the people remaining in the country will not increase their consumption in response to the increased water availability associated with the resettlement. This implies, in particular, that the local per capita water footprint is independent of water availability. We relax this assumption in a robustness check presented in SI, where we instead assume that the per capita water footprint increases linearly with per capita water availability—an assumption that is consistent with available data in many countries (Fig. S8A and Table S8). The outcomes of this analysis (Fig. S7B) are materially similar to our results in Fig. 3B.

The opposite side of the trade-off—the hardship of individual refugees—is challenging to fathom, let alone to represent within an analytical model. Here, we assume that each refugee is waiting for the resolution of hardship, but some longer than others. Within each country $d$, resolutions occur independently at a known average rate $\lambda_d$, so that the waiting time $t$ (hereafter 'individual hardship') follows an exponential distribution across the refugee population (i.e., resolutions within each country of refuge follow a Poisson process),

$$t \sim \exp(\lambda_d). \qquad (10)$$

The rate parameter $\lambda_d$ captures the capacity of the current country of refuge to respond to refugee needs. As researchers, we observe neither $\lambda_d$ nor the refugee's individual hardship $t$, but we assume that with access to case files and better on-the-ground information, the UNHCR is able to identify the individuals with the biggest hardship, and recommends those above some threshold $\bar{t}$ for resettlement. This threshold is identical across countries in order for resettlement to minimize the overall hardship of refugees across countries.

We show in SI that we can infer $\lambda_d$ from the fraction of refugees selected for resettlement in each current country of refuge,

$$\lambda_d = \ln R_d - \ln Q^{\text{UN}}_d \qquad (11)$$

where $Q^{\text{UN}}_d$ and $R_d$ are respectively the number of refugees reallocated from country $d$ under the current resettlement plan (obtained for 2020 from ref. 64), and the total number of registered refugees under the UNHCR mandate in that country (obtained for 2020 from https://www.unhcr.org/refugee-statistics/)).

Once the parameters for the hardship distributions are identified for each current country of refuge, we can compute the remaining individual hardship for any alternative resettlement plans. Formally, let the vector $\mathbf{Q}$ represent a resettlement plan, with each term representing the number $Q_d$ of refugees resettled out of each current country of refuge $d$. For simplicity, we assume that a resettled refugee

no longer experiences hardship and adds no water stress to the destination country. This assumption is based on our data showing resettlement destinations are often higher income with low water stress, such as many countries in the EU (Table 2). We assume that an alternate resettlement plan is feasible if it requires no more hosting capacity than the current plan, i.e., $\sum_d Q_d \leq \sum_d Q_d^{UN}$. Using the approach outlined in SI, we identify the feasible plans that maximize a weighted average between the following two objectives:

(i) The aggregate relief of the individual hardship of refugees. Assuming that hardship is exponentially distributed across the refugees of each current country of refuge $d$, this corresponds to:

$$\Delta \tau(\mathbf{Q}) = \sum_d \int_0^{Q_d} -\frac{1}{\lambda_d} \ln \frac{x}{R_d} dx \qquad (12)$$

(ii) The aggregate relief of the water stress of current countries of refuge:

$$\Delta S(\mathbf{x}) = \sum_d Q_d \Delta s_{d,SR}^{(y)} \qquad (13)$$

Each such plan is Pareto-optimal in the sense that it cannot be simultaneously improved upon for both criteria of the optimization. Fig. 3B shows four candidate plans (the UNHCR plan and alternative plans 2–4), along with the relative relief $\Delta \tau(\mathbf{Q})/\Delta \tau(\mathbf{Q}^{UNHCR})$ and $\Delta S(\mathbf{Q})/\Delta S(\mathbf{Q}^{(plan\ 4)})$. This Pareto-frontier illustrates the tradeoff between relieving the individual hardship of refugees and the collective water stress of the countries of refuge. Water stress relief for three key countries (Jordan, Yemen, and Lebanon) is displayed as bar charts in Fig. 3B for each plan. The baseline water stress values without resettlement (in gray on the bar charts of Fig. 3B) are computed using UNHCR-reported numbers of registered refugees. We exclude Palestinian refugees under the UNRWA mandate and do not account for the supply-side effects of refugee migration on transboundary streamflow into Jordan.

### Food import and water stress relief

The water stress implications of displaced refugees in destination countries are determined by the extent to which the food that they consume has to be sourced domestically (using local water resources) and could be relieved through increased food imports. We evaluate this potential for each food type $k$ in 2016 (the last year of the CWASI dataset), assuming that refugees maintain the dietary habits of their countries of origin.

The quantity of food type $k$ consumed by all refugees in country $d$ is expressed as:

$$C'_{dk} = \sum_o R_{od} c'_{ok}$$

where $R_{od}$ is refugees from country $o$ in country $d$ and $c'_{ok}$ is their per-capita consumption of food type $k$ (obtained from FAO Food Balance Sheets, see SI), assuming that they maintain dietary habits from their country of origin. The fraction $X_{ddk}$ of the virtual water embedded in that food quantity that is produced locally (i.e., *not* imported) contributes to water stress in the country of refuge $d$. This contribution is expressed as:

$$\Delta S_{dk} = \frac{C'_{dk} W_{dk} X_{ddk}}{\epsilon_d (TRWR_d - EFR_d)}$$

where $W_{dk}$ is the virtual water embedded in a tonne of food type $k$ consumed in country $d$ (see SI). The virtual water that would need to be imported into country $d$ to avert this increased water stress can finally be expressed as:

$$\Delta I_{dk} = C'_{dk} \cdot \frac{\sum_p M_{pdk} W_{pk}}{\sum_p M_{pdk}}$$

where $M_{pdk}$ is the virtual water embedded in the (current) import of food type $k$ from country $p$ to $d$ and $W_{pk}$ the virtual water embedded in a tonne of food type $k$ in country $p$. The above expression assumes that the (current) relative contribution of each export country to food type $k$ in country $d$ is maintained. We also assume that the system is 'well-mixed' in the sense that the virtual water volume embedded in a unit of food that is exported from $p$ is identical to that in the food consumed in that country.

Figure 3C displays the water stress relief $\Delta S_{dk}$ ($y$-axis) against the blue water import $\Delta I_{dk}$ (normalized by the total 2016 blue water imports of the country) by host country $d$ and good type $k$.

### Reporting summary

Further information on research design is available in the Nature Portfolio Reporting Summary linked to this article.

## Data availability

The per capita water footprint data generated in this study are available at https://doi.org/10.5281/zenodo.7826777. Refugee data were obtained from the UNHCR data platform https://www.unhcr.org/refugee-statistics/, and the water footprint of primary and processed crops was obtained from the CWASI dataset https://www.watertofood.org/download/. All country-level water metrics were obtained from the FAO AQUASTAT database https://www.fao.org/aquastat/en/.

## Code availability

Data analysis was carried out using R v.4.1.0. and the following packages: fst (v.0.9.8), ggthemes (v.4.2.4), scales (v.1.2.1), weights (v. 1.0.4), plotly (v. 4.10.1), tidyverse (v. 1.3.2), boot (v. 1.3.28), ggplot2 (v. 3.4.0), country code (v. 1.4.0), and maps (v. 3.4.1) all of which are publicly available are no cost at https://www.r-project.org. The code developed to generate the figures is available at https://doi.org/10.5281/zenodo.7826777.

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

## Acknowledgements

We thank Drs. Ilaria Schnyder von Wartensee, Diogo Bolster, Kaveh Madani, Megan Konar, and Rahul Oka for their helpful ideas and feedback. We thank Dr. Jim Yoon for his help on the Jordan case study. We acknowledge support from the National Science Foundation (NSF) awards (G.P., M.C.M.-I., and M.F.M.) ICER 1824951 and EAR 2142967 (L.B. and M.F.M.).

## Author contributions

M.F.M. and M.C.M.-I. designed research; L.B., A.W., M.T., and G.P. performed the analysis; M.F.M. and L.B. wrote the paper.

## Competing interests

The authors declare no competing interests.
