## [Peer Review File · Nature Communications]

Food demand displaced by global refugee migration influences water use in already water stressed countriesReviewer #1 (Remarks to the Author):

The paper discusses an interesting topic, investigating food demand, water stress, and the role of displaced people, looking at Jordan as a case study. Nevertheless, I have the following suggestions:

- I am not clear on what specific literature this paper aims to contribute to; be more explicit about the gap in the literature, on the literature this paper is situated in, and on the research question;
- the issue of the impacts of the war on the Yarmouk river - with decrease in agriculture and increase in the flow towards Al Wanda - has been discussed in previous literature that is not mentioned in your paper, see for instance:

Al Sabeh, Hassan, et al. "Scenario simulation and analysis in the transboundary Yarmouk River basin using a WEAP model." *International Journal of River Basin Management* (2022): 1-22.

Hussein, H. (2017). Whose 'reality'? Discourses and hydropolitics along the Yarmouk River. *Contemporary Levant*, 2(2), 103-115.

- the idea of measuring virtual water and food footprint trade is not original for the case of Jordan; in addition, you need to unpack what the term displaced people and refugees means in the case of Jordan; how do you count them? Numbers on this issue in Jordan are in fact very much contested;
- Coping mechanisms for food security implemented by refugees are not accounted for;
- A major issue is also that the statistics and data look at Jordan only for a determined period, not accounting for the fluctuations over time; it would be more original and interesting to compare these data over time since the 1950s.

Overall, I am not really convinced by the originality of the study, so I would invite the authors to further reflect on this issue.

Reviewer #2 (Remarks to the Author):

Key Findings

The authors make several key findings. They argue that the relationship between refugees and water stress in a host country varies and depends on context and local factors. Importantly, they find that refugees do not significantly contribute to overall water stress in most destination countries. For example, in the EU and some other regional contexts, the presence of refugees has not contributed to water stress. However, this has not been the case for countries like Jordan and Lebanon, where water stress from refugees has been severe, as the authors argue. This finding advances the authors' broader, and compelling, claim that the water footprint of refugee displacement is disproportionately carried by a small number of countries; furthermore, of these countries, it is those that are already at risk of severe water crises that encounter the most severe water stress.

With these findings in mind, the authors make two policy suggestions, supported with empirical data. They argue that policy changes must be made in the short- and long-term to support countries, like

Jordan and Lebanon, that are facing severe water stress. In the short-term, the authors argue that water-abundant countries should export a greater share of water intensive foods to refugee-hosting countries facing water stress in order to lessen local demands on water use. While global trade in of itself cannot substantially relieve water demand or scarcity, as the authors argue, this may be a short-term strategy to mitigate the challenges induced by water stress. In the long-term, the authors propose a revamp to the refugee resettlement system. The authors argue that resettling refugees from water-stressed countries to water-abundant countries in much greater numbers could significantly alleviate water stress. This is a more ambitious policy suggestion, as refugee resettlement remains the least frequent durable solution (by some margin too) in the international refugee regime.

Strengths

The article contributes to an understanding of the local and global repercussions on water stress, which remains to be thoroughly characterized. It does so by arguing, persuasively, that refugees do not significantly contribute to overall water stress in most destination countries, contrary to what might be expected. Rather, the article shows that the water footprint of displacement affects a small group of countries that already face severe water scarcity. This finding is important for a few reasons. First, it may help dispel xenophobic rhetoric that suggests that migrants and refugees always drain the resources of host countries. This finding is especially relevant in the EU context, where discourse on the 'needy' and 'dependent' migrant is on the rise; it is within this context that the authors find that refugees do not necessarily have a significant water footprint in water-abundant countries. Secondly, the article serves as a challenge to overly deterministic perspectives on the climate-migration nexus. Climate determinism and Neo-Malthusian perspectives still appreciate significant backing in the study of 'climate migration'. They maintain that migration necessarily places pressure on the natural environment, thus generating conflict. This article holds, however, that environmental pressures are not a necessary outcome of migration, even mass migration. Rather, greater attention must be paid to underlying structural and contextual factors.

The article is also positioned as a strong critique of inequalities within the international refugee regime. It, importantly, makes the case for overhauling the refugee resettlement system, which is a provocative suggestion given the current state of the regime. It contributes to a growing critical call for countries in the Global North to more actively share the responsibility of refugee accommodation, a responsibility that they have consistently eschewed. That the article highlights inequalities in responsibility sharing and its impact on countries already experiencing severe water scarcity is a notable contribution.

Suggestions

The article could benefit from more thoroughly grounding the discussion of water scarcity and responsibility gaps in the language of inequality. While the authors do mention the disproportionate burden placed on host countries facing water scarcity, we do not quite get a sense of why these countries are water scarce in the first place, nor the factors (legacies of colonialism, climate change, economic inequalities, resource mismanagement, etc.) that exacerbate it. The article does not address the politics or origins of scarcity. While this may fall outside the scope of an article about the water footprint of refugees in host countries, how can we conceptualize ways to mitigate environmental pressures (water or otherwise) without considering the roots of scarcity? There is a way to do this that situates the article in a greater discussion of global inequalities, whether in the international refugee regime, in the differential impacts of climate change, or in inequalities in the global economy.

Secondly, the article would benefit from some insight, or at least a call for future research, on how internally displaced persons fit into this discussion. The reader is given no information about the relationship between IDPs and their water footprints other than the authors' note that IDPs "do not cause direct international water demand displacement through migration." There is no corresponding citation for this claim, which underscores a need to address, even briefly, if and how the water stress

implications differ between IDPs and refugees. Certainly, this issue deserves more attention than just a sentence and a half towards the end of the manuscript, especially so given that the authors acknowledge that global internal displacement numbers far surpass instances of cross-border forced migration.

Finally, I find the organization of the article to be unusual, but this might be due to differing disciplinary conventions.

Reviewer #3 (Remarks to the Author):

Dear Authors,

Thanks for this impressive manuscript. You will see that my review (attached) is quite critical. Having worked on water and migration issues, I feel that the paper's approach to this very complex issue is reductionist and potentially misleading. I would invite the authors to consider the application of their framework to migration policy in general and discuss it with a migration expert, if possible.

The paper *Displaced food demand and the uneven water stress implications of global refugee migrations* presents a quantification of the impact of refugee movement on the demand for water. For disclosure, I had the opportunity to review an earlier version of this manuscript (earlier review in Annex 1). I am thankful to the authors for having considered my comments, many of which seem to have been adopted in this revised version of the paper.

My feeling remains the same: I commend the paper's focus on the important topic of water and migration and its solid and comprehensive data analysis efforts; however, I think it addresses a very complex issue with a reductionist framework that focuses on volumetric water stress considerations. In addition, the paper's finding that millions of Syrians have put massive pressure on Jordan's natural resources is not novel.

MAJOR COMMENTS

1. **Reductionist framework and need to further tone-down results.** Water stress is a crude volumetric indicator of a country's water issues. The authors could consider further toning down their title and abstract to acknowledge that their paper adopts a reductionist approach to study a very complex problem. The abstract goes some way in this direction, but I think it should adopt more cautious language.
2. **Explain novelty of these findings to Jordanian policy-makers.** A reader familiar with the Middle East and Jordan will not be surprised to find out that millions of refugees exacerbated water issues in Jordan, which already had severe water issues even before the Syrian crisis (e.g., non-revenue water, a highly precarious financial situation of the water sector due to escalating electricity costs). While the authors note this in some parts of the manuscript, this should be front and centre. The authors are calculating the blue water footprint of food demand from refugees – this is elegant and gives a new quantitative estimate but does not really provide any new policy insight to Jordanian policy-makers in my opinion. A recent policy report from the Ministry of Water and Irrigation¹ came to the same conclusion but at least provided a much more balanced view of reality on the ground and considered a wider range of policy options.
3. **Policy alternatives and decision to focus on refugee resettlement policies.** Tony Allan came-up with the notion of virtual water trade to explain how countries in the Middle East continue to grow economies and populations despite water scarcity (Allan 2012) [Please cite Tony Allan – without his contribution we would not be discussing this paper – and also have a look at the recent collection of papers published in *Water International* discussing his contribution]. The authors use this framing to consider virtual water trade as one of the policy levers that Jordan can deploy to deal with refugee-linked water scarcity. This is good. I suggest stopping there so as to remain within the narrow 'volumetric' framing of the paper and not enter refugee resettlement policy. If the authors want to keep the refugee resettlement policy trade-off analysis, then they should at least explain why they single it out among additional policy alternatives. This might include, for example, investments in hydroponic farming to give refugees the opportunity to grow their own food without consuming water and land resources (Verner et al. 2017) This is just an example of

¹ Influx of Syrian Refugees in Jordan | Effects on the Water Sector. Available from: <https://reliefweb.int/report/jordan/influx-syrian-refugees-jordan-effects-water-sector>

the wide portfolio of options not considered in the paper. I suggest the policy analysis section to just focus on virtual water calculations.

EARLIER COMMENTS WHICH I THINK WERE NOT CONSIDERED, GRATEFUL IF THE AUTHORS COULD PROVIDE ANSWERS TO SHED LIGHT ON SOME ASSUMPTIONS AND IMPLICATIONS (ESPECIALLY #2).

1. **Implicit assumption that water released by resettlement will not be used in Jordan.** By suggesting that refugees hosted in Jordan should be resettled somewhere else to relieve water stress, the paper assumes that the water 'released' by resettling refugees will not be used by other water users in Jordan. The authors should clarify if they are implicitly assuming that the water released by resettling refugees outside of Jordan would not be used to meet other uses in Jordan (thus negating any reductions in water stress).
2. **Implications for migration policy more in general.** I think it is useful to take a step back and consider the paper's implications for migration policy more in general. In other words, let's consider the paper's implications for economic migrants, who account for a much larger share of overall population movement. According to the paper's framing (if I understand things correctly), population movement transfers blue water demand towards countries with water-intensive dietary habits. This effect is obviously more pronounced in countries with already high blue water stress. Hence, if we blindly apply the paper's approach, we should suggest to Qatar and UAE (two countries with high levels of freshwater stress) to reduce their very large populations of economic migrants because they increase blue water stress. I think we would tend to agree that such policy recommendation would be misplaced, not least because it ignores the migrants' contribution to these countries' economic productivity and wealth. Can the authors comment on this point, which in my opinion demonstrates the rather reductionist approach taken in the paper?

MINOR POINTS

- Line 74-75: this seems to be factually incorrect given current refugee numbers : Venezuela, Ukraine and South Sudan are among the top 5 countries of origin of refugees, and they are all water rich countries (and so are the host countries). Please consider revising this statement and providing the years it refers to.

CONCLUDING REMARKS

Please note again that I am very impressed by the paper's ambition and rigorous analytical approach. This work is of high quality. However, having worked on migration issues myself, I just want to really advise the authors to keep the paper on a defined, narrow track and be as humble as possible in presenting findings.

References

Allan, T. (2012). *The Middle East water question: Hydropolitics and the global economy*. Bloomsbury Publishing.

Verner, D., Vellani, S., Klausen, A. L., & Tebaldi, E. (2017). *Frontier Agriculture for Improving Refugee Livelihoods*. World Bank: Washington, D.C.

Review of The uneven water stress implications of global refugee migrations

The paper 'The uneven water stress implications of global refugee migrations' presents the first-ever application of the concept of virtual water to migration policy. I respect the authors' work and effort: the data and analytical approach are commendable, as is the paper's objective to study the timely topic of forced displacement and its impact on host countries' water security. However, I feel that the manuscript takes on a complex issue with a fairly simplistic conceptual framework based on virtual water. In short: while I commend the paper's focus on the important topic of water and migration, I believe that the conceptual approach used to study it is inadequate and misleading. The fundamental conceptual flaw is that the paper assumes that migration policy decisions should be guided only by considerations of trade-offs between water stress estimated based on virtual water and hardship relief.

Building on this overarching comment, I would like to invite the authors to unpack some of their assumptions.

- **Implicit assumption that water released by resettlement will not be used in Jordan.** First, by suggesting that refugees hosted in Jordan should be resettled somewhere else to relieve water stress, the paper assumes that the water 'released' by resettling refugees will not be used by other water users in Jordan. The authors should clarify if they are implicitly assuming that the water released by resettling refugees outside of Jordan would not be used to meet other uses in Jordan (thus negating any reductions in water stress).
- **Greywater (Wastewater) reuse is significant, especially in Jordan.** If I understand the paper correctly, the data used does not include grey water. I understand grey water to mean wastewater, which is a key supply source for irrigation in many arid countries. In Jordan, for example, about 25% of irrigation water demands are met through wastewater (United Nations 2022). Failure to account for such an important water source is likely to significantly skew the results. Is there a way to control for the role of wastewater in meeting water demand for food production?
- **Opportunity cost of water.** As you know, scholars have criticized the concept of virtual water because of its inability to consider the opportunity cost of water (Wichelns 2004). In conditions of scarcity, the objective of water management should not be to minimize use of water resources per se, but rather to ensure that the value of water for society and the environment is maximized. I think the paper implicitly assumes that water policy in Jordan should minimize water use irrespective of the value of water to users, and I invite the authors to clarify this point and/or make this assumption explicit. The authors need to acknowledge the limitations of virtual water and reflect its inability to represent the opportunity cost of water, especially if their ambition is to inform practical water policy decisions.
- **Changing diets and the assumptions underlying short-run and long-run blue water requirements.** The references supporting assumptions about the dietary choices of refugees in the short and long run do not provide a full global coverage (i.e., for example no references for refugees from Latin America or Middle East). I invite the authors to provide additional references

and further justify their central assumption that dietary patterns shift over the long term. Ideally this would be based on data or a more thorough literature review.

- **Why should water stress be singled out when devising migration policy?** The paper only studies the broad water-migration nexus through the lens of virtual water and its role in influencing water stress. This means, in my opinion, that the authors fail to address a highly relevant question: why should water be singled out when advising UNHCR on resettlement plans? Refugees are not just consumers of water resources, but also valuable providers of labour (especially in agriculture), for example. They can also contribute to steep increases in prices of consumer goods and real estate, benefiting some parts of the economy. Similarly, with more workers in the labour market, wages can increase, thus helping employers but hurting employees (e.g., Alix-Garcia et al. 2017). A sound migration and resettlement policy should take these factors into account before jumping to conclusions. While there might be no space in this paper quantify/model all these factors, I suggest that the authors try to at least illustrate how these factors interact with the virtual water element using a conceptual framework (see Alix-Garcia et al. 2017 for an example of a framework for studying the impact of refugees on host communities). This will help ringfence recommendations and show readers the limited scope of the paper.
- **Illustrative analysis or policy actionable recommendations?** The first sentence of the conclusions suggests that the authors are cognizant of the ‘illustrative’ nature of their analysis; however, the abstract and title are much bolder and more direct, giving readers the false impression that the paper’s findings could be used to actually justify resettlement decisions. I invite the authors to reflect on the breadth of the policy question they are attempting to address and on the actionability of their recommendations. Did the paper consider a wide-portfolio of potential solutions to address water stress in host communities? Are the authors confident to say that resettlement based on virtual water analysis is the best approach to manage the environmental pressures posed by refugees? Instead of investing money in resettling refugees, why not invest in hydroponic farming to give refugees the opportunity to grow their own food without consuming water and land resources? (Verner et al. 2017) This is just an example of the wide portfolio of options not considered in the paper because of the narrow focus on virtual water.
- **Implications for migration policy more in general.** I think it is useful to take a step back and consider the paper’s implications for migration policy more in general. In other words, let’s consider the paper’s implications for economic migrants, who account for a much larger share of overall population movement. According to the paper’s framing (if I understand things correctly), population movement transfers blue water demand towards countries with water-intensive dietary habits. This effect is obviously more pronounced in countries with already high blue water stress. Hence, if we blindly apply the paper’s approach, we should suggest to Qatar and UAE (two countries with high levels of freshwater stress) to reduce their very large populations of economic migrants because they increase blue water stress. I think we would tend to agree that such policy recommendation would be misplaced, not least because it ignores the migrants’ contribution to these countries’ economic productivity and wealth.
- **Articulation of the paper’s conceptual contribution.** I would invite the authors to better articulate their conceptual contribution to our understanding of water and migration. In my opinion, the underlying conceptual contribution is weak and largely based on the extension of virtual water to the case of population movement. As noted in Nature Sustainability’s editorial from August 2021 (Too much and not enough), water research is stagnating and there is a need

to push the field forward through conceptual advances and theoretical innovations and not just new application of existing concepts (e.g., footprint analyses of embodied water/virtual water). Hence, I invite the authors to more openly discuss their perceived contribution beyond the application.

- **To conclude, the complexity of the issue requires a careful approach and a much humbler presentation of the results.** Refugee arrivals have both positive and negative effects on host communities, and a resettlement policy based only on the objective of reducing water stress (especially if quantified only through virtual water) is, in my opinion, risky. Migration policy is a highly charged political issue, so publishing this paper in a high-impact journal such as *Nature Sustainability* could lead to severe unintended consequences. The authors themselves recognize that the analysis is only 'illustrative'; however, the paper's title, abstract and results sections give readers the perception that findings are policy-relevant and actionable which, in my opinion, is not the case.

References

Alix-Garcia, J., Artuc, E., & Onder, H. (2017). *The Economics of Hosting Refugees*. World Bank, Washington, DC.

United Nations (2022) Policy brief: Decentralized Wastewater Treatment Systems (DWATS) as a Climate Change Adaptation Option for Agriculture in Jordan. UN Jordan. Available from: https://jordan.un.org/sites/default/files/2022-03/EN_Policy_Brief_DWATs.pdf

Verner, D., Vellani, S., Klausen, A. L., & Tebaldi, E. (2017). *Frontier Agriculture for Improving Refugee Livelihoods*. World Bank: Washington, D.C.

Wichelns, D. (2004). The policy relevance of virtual water can be enhanced by considering comparative advantages. *Agricultural Water Management*, 66(1), 49-63.

Reviewer #1

The paper discusses an interesting topic, investigating food demand, water stress, and the role of displaced people, looking at Jordan as a case study. Nevertheless, I have the following suggestions:

1.1. I am not clear on what specific literature this paper aims to contribute to; be more explicit about the gap in the literature, on the literature this paper is situated in, and on the research question;

Thanks for taking the time to review our paper. We believe that our paper makes novel contribution to two distinct bodies of literature:

First, we extend the concept of virtual water flows to migration (displaced people), whereas previous work has almost exclusively applied the framework to international trade networks (displaced goods). Only one previous study that we are aware of (Ref 22 in the manuscript) evaluated the water footprint of migrants, but it was focused on economic migration and its effect on international trade fluxes (i.e. displaced goods). This manuscript is the first one to characterize virtual water fluxes associated with the physical displacement of food demand through migration. This 'demand-side' network of virtual water flows contrasts sharply with the 'supply-side' flows documented for international trade. We find that these 'demand-side' flows of virtual water are an order of magnitude smaller than the 'supply-side' flows associated with trade, but they are not insignificant. In particular, in contrast to trade, they do not tend to "equilibrate" heterogeneities in blue water availability but often drive virtual water away from some of the world's most arid regions.

Second, we enrich the literature on the association between water and armed conflicts. While a lot has been written about the water-related drivers of armed conflict, much less has been written about the water-related impacts of warfare. What little work has been done has generally focused on the impact of conflicts on local waters, e.g., through pollution. Recent work (Ref 9) showed that conflict can influence water resources beyond the battlefield along hydrological features. Here, for the first time, we document how local conflicts can impact water resources over much larger distances by displacing the water demand associated with migrants.

These two contributions are clarified in the revised introduction as:

L54 (Introduction)

This effort contributes to two distinct bodies of literature. The first contribution relates to the relationship between water resources and armed conflicts. Most recent research focuses on water (or lack thereof) as a {driver}, rather than consequence, of armed conflicts (Shillinger et al, 2020). While an increasing body of work evaluates the impact of warfare on the quality and quantity of local waters (Francis et al, 2011), recent work in

Syria showed that conflict-related changes in land use can affect regional water availability beyond the battlefield (Muller et al., 2016). Here, we show that the hydrological impacts of localized conflicts can be truly global, through the effect of trade and migration. Our second contribution is to extend the concept of virtual water from its original application to supply chains and the water footprint of displaced goods, to migrations and the water demand of displaced people. By keeping track of the ultimate origin of the water embedded in traded goods, we elucidate the direct (migration) and indirect (trade) effect of refugee displacement on country-level water stress. The new water footprint dataset that we present captures these effects by uniquely distinguishing the effects of dietary habits, globalized supply chains, and agricultural water use efficiency.

We believe that the manuscript also makes helpful contributions within the specific context of Jordan, which we discuss in response to your comment 1.3 below.

1.2. The issue of the impacts of the war on the Yarmouk river - with decrease in agriculture and increase in the flow towards Al Wanda - has been discussed in previous literature that is not mentioned in your paper, see for instance. Al Sabeh, Hassan, et al. "Scenario simulation and analysis in the transboundary Yarmouk River basin using a WEAP model." *International Journal of River Basin Management* (2022): 1-22.; Hussein, H. (2017). Whose 'reality'? Discourses and hydropolitics along the Yarmouk River. *Contemporary Levant*, 2(2), 103-115.

Thanks, we have added these references to the manuscript on L 188 and 190

1.3. the idea of measuring virtual water and food footprint trade is not original for the case of Jordan;

We agree that the challenges faced by the Jordanian water-sector have been well documented and have added appropriate references in the manuscript (see copy-pasted text below). These challenges include its reliance on non-renewable groundwater, internationally shared resources (and associated tensions), dependance on food imports and growing demand due to large influxes of refugees. However, to our knowledge, only one previous study estimates and discusses the country's water footprint and reliance on imported virtual water within a systematic quantitative framework (Shyns et al 2015, now Ref 46 in the revised manuscript). This analysis uses dated (2005) data and is a country-level analysis that does not resolve refugee migration as an important source of (outgoing) virtual water flux in addition to (incoming) virtual water flux associated with food imports. Our analysis fills this gap by using more recent footprint data to specifically estimate the virtual water embedded in refugee migration fluxes and their effect on the overall water footprint of the country, which turns out to be significant in Jordan. .

While our analysis is primarily meant to fill the gaps in the broader scholarly literature on conflict and virtual water networks that we discuss in response to your first comment, we do believe that our results are also practically relevant for the specific context of Jordan. We caution multiple times in the manuscript (e.g., Abstract and L. 263) that water footprint estimates and virtual water flows are crude metrics that should not, alone, be used to inform policy. However, they provide important aggregate-level quantitative information to support more elaborate models of the Jordanian water sector (e.g., Ref 10) that can be used to generate specific policy recommendations.

L175 (Regional Impacts)

The multi-faceted challenges faced by the Jordanian water sector are well documented (Yoon et al, 2021, Syhns et al 2015, Hussein et al 2017, Courcier et al 2005, Whitman et al 2019, Mustafa et al 2019, Al Sabeh et al, 2022), and include the need to supply water to an increasing population of refugees (Breulmann et al, 2021). Unlike previous efforts, this study documents the **food** demand displaced by refugee displacements into Jordan, which we estimate has contributed 47 to 75 percentage points to the country's overall water stress (Table 2, JOR short and long run).

L272 (Policy Implications)

Finally, we also believe that a quantitative estimate of the virtual water fluxes associated with forced population displacement, as provided here for the first time, is helpful to inform more complex operational models to support policy (see e.g., Yoon et al, 2021 for Jordan).

1.4. in addition, you need to unpack what the term displaced people and refugees means in the case of Jordan; how do you count them? Numbers on this issue in Jordan are in fact very much contested;

Thank you -- we are aware that this is a contentious issue and so have tried to be as clear as possible in the manuscript on the origins of these numbers.

The results section that first discusses Jordan now reads:

L168 (Regional Impacts)

While specific numbers vary across sources (see Methods and Table 2), approximately 1.2 and 1.1 million Syrians had respectively crossed into Lebanon and Jordan (respectively) by 2016 (Yoon et al., 2022; Jafaar et al 2020).

The caption to Table 2 reads:

LBN and JOR include estimates of Syrian refugees (registered and unregistered) obtained from (Yoon et al., 2022; Jafaar et al 2020) for Lebanon and Jordan, respectively. LBN and JOR also include registered refugees from all the other countries in the UNHCR dataset for 2016, which includes Palestinians under UNRWA mandate. JOR* only includes registered and unregistered Syrian refugees from (Yoon et al, 2021)

and estimates the net effect of the Syrian refugee crisis on Jordanian water stress by subtracting the exploitable increase in transboundary river flow (18 MCM/y) from Δ BWF when determining Δ Stress.

The revised Methods Section reads:

L305

The dataset includes international refugees that seek refuge outside of their country of origin and are included under UNHCR or UNRWA mandates. Unless otherwise noted, we exclude migrants that are not registered with a UN agency. Approximately 1.1 and 1.2 million Syrian refugees have respectively been reported in Jordan (Yoon et al., 2022) and Lebanon (Jafaar et al, 2020) in 2016, of whom 0.65 and 1.0 million (respectively) are registered and included in UNHCR data. For these two countries in particular, we included unregistered refugees in country-specific results when indicated (i.e. Table 2 and Figure 3).

The caption to Figure 3 reads:

Note that the analyses in Panels **B** and **C** only include registered refugees under UNCHR mandate, which are eligible to be considered for resettlement.

We also now explicitly discuss Internally Displaced Persons (IDP) in the revised Methods section:

L310

We focus on international refugees for consistency with the virtual water data set that provides water footprint estimates at the country level. As discussed in the main text, this scale of analysis relies on the premise of a working domestic food supply system that conveys food from the agricultural production regions of the country to the marginal lands where refugees often hosted (Jacobsen et al, 1997). This assumption generally does not hold for internally displaced persons (IDP) that are displaced within their own war-torn country. IDPs outnumber international refugees by nearly two to one (UNHCR2017) but had to be excluded from our analysis due to unavailable water footprint data at the sub-national scale. Recent research using satellite imagery finds evidence of enhanced land use change in the vicinity of IDP camps (Lang et al, 2010; Kranz et al 2015), suggesting a sizable local impact on food and water availability that remains to be characterized.

1.5. Coping mechanisms for food security implemented by refugees are not accounted for;

This is an important caveat that relates to our assumption that refugees preserve the dietary habits from their country of origin. While this assumption is supported by ethnographic research (see Refs 78 and 84), it might not hold in strongly food insecure conditions. We have modified the Methods section to clarify this point.

L383, Methods

Our assumption that recent refugees preserve their dietary habits from their country of origin is broadly supported by ethnographic evidence. For example, research in refugee camps in Kenya (Oka 2014) and Rwanda (Taylor et al. 2016) suggest that nearly all refugees sell part or most of their food allotment on the black market to buy food that conforms with their traditional dietary habits (thus creating a sense of normalcy (Oka 2014)). However, it is important to note that the assumption neglects any potential coping mechanisms to address prevailing conditions of food insecurity (Nisbet et al, 2022). Neglecting these coping mechanisms (e.g., eating less or less diverse (Olaimat et al, 2022)) might cause us to overestimate the BWF of displaced refugees in food-insecure conditions.

1.6. A major issue is also that the statistics and data look at Jordan only for a determined period, not accounting for the fluctuations over time; it would be more original and interesting to compare these data over time since the 1950s.

We agree that a full historical analysis of Yarmouk river flows (and the Jordanian water sector as a whole) would be interesting, but would unfortunately go beyond the scope of the paper, which is a global analysis of contemporaneous refugee migrations. The focus on Jordan in this paper is limited to determining the net impact of Syrian refugee displacement on water availability. For the specific analysis of the Yarmouk (which we believe the reviewer refers to), we believe that annual flows observed before the Syrian conflict are the appropriate baseline. We make sure to clarify in the revised manuscript that this “baseline” flow follows a long history of decreasing flows, and refer to the specific literature (Ref 47) where these long term patterns are discussed. We also added a supplementary Figure showing annual Yarmouk flows at Al Wehda dating back to the early 80's.

L187, Regional Impacts

On the supply side, average Yarmouk river flow from Syria into Jordan has increased by about 50 MCM/yr between 2011 and 2015 (compared to pre-2011 levels, Figure 3A, dashed), a sudden increase that occurred on the heels of decades of steadily decreasing flows (Figure S7) and a history of water allocation disputes between Jordan and Syria (Hussein, 2017).

1.7. Overall, I am not really convinced by the originality of the study, so I would invite the authors to further reflect on this issue.

Thank you for your feedback and we hope that our answers to your comments (particularly 1.1. and 1.3) makes a convincing case of the novelty of our findings.

Reviewer #2

2.1. Key Findings: The authors make several key findings. They argue that the relationship between refugees and water stress in a host country varies and depends on context and local factors. Importantly, they find that refugees do not significantly contribute to overall water stress in most destination countries. For example, in the EU and some other regional contexts, the presence of refugees has not contributed to water stress. However, this has not been the case for countries like Jordan and Lebanon, where water stress from refugees has been severe, as the authors argue. This finding advances the authors' broader, and compelling, claim that the water footprint of refugee displacement is disproportionately carried by a small number of countries; furthermore, of these countries, it is those that are already at risk of severe water crises that encounter the most severe water stress. With these findings in mind, the authors make two policy suggestions, supported with empirical data. They argue that policy changes must be made in the short- and long-term to support countries, like Jordan and Lebanon, that are facing severe water stress. In the short-term, the authors argue that water-abundant countries should export a greater share of water intensive foods to refugee-hosting countries facing water stress in order to lessen local demands on water use. While global trade in of itself cannot substantially relieve water demand or scarcity, as the authors argue, this may be a short-term strategy to mitigate the challenges induced by water stress. In the long-term, the authors propose a revamp to the refugee resettlement system. The authors argue that resettling refugees from water-stressed countries to water-abundant countries in much greater numbers could significantly alleviate water stress. This is a more ambitious policy suggestion, as refugee resettlement remains the least frequent durable solution (by some margin too) in the international refugee regime.

Thank you for taking the time to review our paper -- this is a great summary of our findings. We agree that resettlement or repatriation remain low frequency durable solutions in the international refugee regime. We present them as aspirational goals in terms simultaneously providing a lasting solution for the refugees themselves, while relieving the often resource-poor countries that currently shelter them. The paper finishes with a call for action for the international community to step up resettlement efforts:

L287 Policy Implications

Resettlement can simultaneously provide a lasting solution for the refugees themselves, while relieving the often resource-poor countries that currently shelter them. Our analysis illustrates the dynamics of this win-win proposition in the specific context of water stress. Yet, as of 2021, less than 3\% of the 1.44 million refugees of the UNHCR plan have been successfully relocated (UN 2021). There is an urgent need -- and an ethical imperative -- for higher-income countries to support the resettlement effort through increased resettlement pledges.

2.2. Strengths: The article contributes to an understanding of the local and global repercussions on water stress, which remains to be thoroughly characterized. It does so by arguing, persuasively, that refugees do not significantly contribute to overall water stress in most destination countries, contrary to what might be expected. Rather, the article shows that the water footprint of displacement affects a small group of countries that already face severe water scarcity. This finding is important for a few reasons. First, it may help dispel xenophobic rhetoric that suggests that migrants and refugees always drain the resources of host countries. This finding is especially relevant in the EU context, where discourse on the 'needy' and 'dependent' migrant is on the rise; it is within this context that the authors find that refugees do not necessarily have a significant water footprint in water-abundant countries. Secondly, the article serves as a challenge to overly deterministic perspectives on the climate-migration nexus. Climate determinism and Neo-Malthusian perspectives still appreciate significant backing in the study of 'climate migration'. They maintain that migration necessarily places pressure on the natural environment, thus generating conflict. This article holds, however, that environmental pressures are not a necessary outcome of migration, even mass migration. Rather, greater attention must be paid to underlying structural and contextual factors. The article is also positioned as a strong critique of inequalities within the international refugee regime. It, importantly, makes the case for overhauling the refugee resettlement system, which is a provocative suggestion given the current state of the regime. It contributes to a growing critical call for countries in the Global North to more actively share the responsibility of refugee accommodation, a responsibility that they have consistently eschewed. That the article highlights inequalities in responsibility sharing and its impact on countries already experiencing severe water scarcity is a notable contribution.

We are extremely grateful to the reviewer for their input, which is absolutely on point and helped us put our results in the perspective of the broader societal conversation around migration and climate. We have distilled some of the key points in the reviewer's (excellent synthesis) in our revised manuscript, for example:

L275 (Policy Implications)

Reassuringly, our results have shown that water stress associated with increased food consumption is not a prominent issue in the overwhelming majority of destination countries {This finding is important in the context of increasing rhetoric from the far-right that present migrants and refugees as drains the resources of host countries (see, e.g., (Beck et al, 2017) for the European Union)

L209 Refugee Resettlement and Food Import

Our analysis has shown that the water stress of refugee displacements is disproportionately borne by a few vulnerable countries that host an incommensurate number of refugees amidst severe preexisting water security challenges. This burden exacerbates a broader landscape of water inequality and injustice in the Global South, where resource mismanagement (Gleick 2014), poor governance (de Chatel 2014, Lu et al, 2014), the commodification of water and land (d'Odorico et al, 2017), extractive industries (Chiarelli et al, 2022), environmental cost shifting (Dell'Angelo et al, 2018) and

the legacies of colonialism (Gasteyer et al 2012) often amplify water scarcity in arid and semi-arid regions that will be strongly impacted by climate change (Falkenmark et al, 2013)

2.3. The article could benefit from more thoroughly grounding the discussion of water scarcity and responsibility gaps in the language of inequality. While the authors do mention the disproportionate burden placed on host countries facing water scarcity, we do not quite get a sense of why these countries are water scarce in the first place, nor the factors (legacies of colonialism, climate change, economic inequalities, resource mismanagement, etc.) that exacerbate it. The article does not address the politics or origins of scarcity. While this may fall outside the scope of an article about the water footprint of refugees in host countries, how can we conceptualize ways to mitigate environmental pressures (water or otherwise) without considering the roots of scarcity? There is a way to do this that situates the article in a greater discussion of global inequalities, whether in the international refugee regime, in the differential impacts of climate change, or in inequalities in the global economy.

We wholeheartedly agree. We have added a short discussion to the manuscript that clearly links it to the broader scholarly discussion on global inequalities with a particular focus on water justice. We are aware that the limited scope and length of the manuscript does not do justice to the complexity of the topic, but hope that the added references are enough to situate the paper within that literature.

L209 Refugee Resettlement and Food Import

Our analysis has shown that the water stress of refugee displacements is disproportionately borne by a few vulnerable countries that host an incommensurate number of refugees amidst severe preexisting water security challenges. This burden exacerbates a broader landscape of water inequality and injustice in the Global South, where resource mismanagement (Gleick 2014), poor governance (de Chatel 2014, Lu et al, 2014), the commodification of water and land (d’Odorico et al, 2017), extractive industries (Chiarelli et al, 2022), environmental cost shifting (Dell’Angelo et al, 2018) and the legacies of colonialism (Gasteyer et al 2012) often amplify water scarcity in arid and semi-arid regions that will be strongly impacted by climate change (Falkenmark et al, 2013)

L213 Refugee Resettlement and Food Import

We recognize that mitigating these issues is a complex undertaking that requires appreciation for the ‘relational, situated, and context-sensitive rather than universalistic’ nature of water justice (Zwarteveen et al., 2014; Sultana 2018), and that focusing on the water stress of refugee displacements is inevitably a partial -- and somewhat reductionist -- perspective. With these caveats in mind, we investigate the potential to alleviate this increased water burden,} either by transferring virtual water demand \emph{away} from vulnerable host countries using refugee resettlement programs; or by transferring virtual water supply \emph{towards} vulnerable countries through enhanced food imports.

2.4. Secondly, the article would benefit from some insight, or at least a call for future research, on how internally displaced persons fit into this discussion. The reader is given no information about the relationship between IDPs and their water footprints other than the authors' note that IDPs "do not cause direct international water demand displacement through migration." There is no corresponding citation for this claim, which underscores a need to address, even briefly, if and how the water stress implications differ between IDPs and refugees. Certainly, this issue deserves more attention than just a sentence and a half towards the end of the manuscript, especially so given that the authors acknowledge that global internal displacement numbers far surpass instances of cross-border forced migration.

This is an excellent point! We added a paragraph in the methods section discussing IDPs. We describe why they had to be excluded from the current analysis (lack of subnational water footprint data) and call for future research to fill that gap.

L310 Methods

We focus on international refugees for consistency with the virtual water data set that provides water footprint estimates at the country level. As discussed in the main text, this scale of analysis relies on the premise of a working domestic food supply system that conveys food from the agricultural production regions of the country to the marginal lands where refugees often hosted (Jacobsen et al, 1997). This assumption generally does not hold for internally displaced persons (IDP) that are displaced within their own war-torn country. IDPs outnumber international refugees by nearly two to one \cite{UNHCR2017} but had to be excluded from our analysis due to unavailable water footprint data at the sub-national scale. Recent research using satellite imagery finds evidence of enhanced land use change in the vicinity of IDP camps (Lang et al, 2010; Kranz et al 2015), suggesting a sizable local impact on food and water availability that remains to be characterized.

2.5. Finally, I find the organization of the article to be unusual, but this might be due to differing disciplinary conventions.

Yes, having methods in a separate section after the main text is a requirement of the journal.

Reviewer #3

3.1. Dear Authors, Thanks for this impressive manuscript. You will see that my review (attached) is quite critical. Having worked on water and migration issues, I feel that the paper's approach to this very complex issue is reductionist and potentially misleading. I would invite the authors to consider the application of their framework to migration policy in general and discuss it with a migration expert, if possible. The paper Displaced food demand and the uneven water stress implications of global refugee migrations presents a quantification of the impact of refugee movement on the demand for water. For disclosure, I had the opportunity to review an earlier version of this manuscript (earlier review in Annex 1). I am thankful to the authors for having considered my comments, many of which seem to have been adopted in this revised version of the paper. My feeling remains the same: I commend the paper's focus on the important topic of water and migration and its solid and comprehensive data analysis efforts; however, I think it addresses a very complex issue with a reductionist framework that focuses on volumetric water stress considerations.

Thanks for agreeing to re-review our paper and we are glad that we have successfully addressed some of your concerns. We agree that this is a complex subject that is at risk of being oversimplified. We went through a thorough review of the manuscript (in consultation with a migration expert on the Notre Dame campus) to make sure the language is cautious enough to mitigate that risk. We have also added explicit discussions on the complexity of both water and migration issues and the limited scope of our analysis (see our responses to your comments 3.3, 3.6. and 3.8 below).

3.2. In addition, the paper's finding that millions of Syrians have put massive pressure on Jordan's natural resources is not novel.

Thank you. While we agree that the fact that millions of Syrian refugees have put pressure on Jordan's natural resources is hardly novel, our analysis is the first (to our knowledge) to provide a *quantitative* estimate of the water needed to feed these refugees and its overall effect on Jordan's aggregate water stress. Our revised results section reads:

L175 (Regional Impacts)

The multi-faceted challenges faced by the Jordanian water sector are well documented (Yoon et al, 2021, Syhns et al 2015, Hussein et al 2017, Courcier et al 2005, Whitman et al 2019, Mustafa et al 2019, Al Sabeh et al, 2022), and include the need to supply water to an increasing population of refugees (Breulmann et al, 2021). Unlike previous efforts, this study documents the *food* demand displaced by refugee displacements into Jordan, which we estimate has contributed 47 to 75 percentage points to the country's overall water stress (Table 2, JOR short and long run).

In addition, this paper puts into context previous findings that refugee out-migration from Syria and the subsequent abandonment of irrigated agriculture has increased transboundary freshwater flows into Jordan through the Yarmouk river (Ref 9 in the manuscript). We compare these effects explicitly in Figure 3A and find that the increase in water demand from in-migrating refugees substantially exceeds the observed increase in streamflow. The relevant results section of the manuscript reads:

L187 (Regional Impacts)

On the supply side, average Yarmouk river flow from Syria into Jordan has increased by about 50 MCM/yr between 2011 and 2015 (compared to pre-2011 levels, Figure 3A, dashed), a sudden increase that occurred on the heels of decades of steadily decreasing flows (Figure S7) and a history of water allocation disputes between Jordan and Syria (Hussein 2018). A recent estimate suggests that approximately 55% of the observed flow increase can be attributed to abandoned irrigation agriculture in upstream Syria by fleeing refugees (Muller et al., 2016; Al Sabeh et al, 2017) (Figure 3A, dotted). Based on a comprehensive model of the Jordanian water sector (Yoon et al, 2021), we estimate that approximately 80% (or 18 MCM) of the Yarmouk increase attributed to refugees was used by Jordanian agriculture in 2015 (see SI). This increase in blue water supply only offsets about 39% of the blue water demand associated with registered Syrian refugees in Jordan (Figure 3A, solid line vs. teal area).

The paper also fills broader gaps in the literatures associated with (i) the bidirectional relationship between water and conflict and (ii) the nature of international virtual water fluxes, as discussed in the revised manuscript (see response to comment 1.1 above). The revised introduction reads:

L54 (Introduction)

This effort contributes to two distinct bodies of literature. The first contribution relates to the relationship between water resources and armed conflicts. Most recent research focuses on water (or lack thereof) as a {driver}, rather than consequence, of armed conflicts (Schillinger et al, 2020). While an increasing body of work evaluates the impact of warfare on the quality and quantity of local waters (Francis et al, 2011), recent work in Syria showed that conflict-related changes in land use can affect regional water availability beyond the battlefield (Muller et al., 2016). Here, we show that the hydrological impacts of localized conflicts can be truly global, through the effect of trade and migration. Our second contribution is to extend the concept of virtual water from its original application to supply chains and the water footprint of displaced goods, to migrations and the water demand of displaced people. By keeping track of the ultimate origin of the water embedded in traded goods, we elucidate the direct (migration) and indirect (trade) effect of refugee displacement on country-level water stress. The new water footprint dataset that we present captures these effects by uniquely distinguishing the effects of dietary habits, globalized supply chains, and agricultural water use efficiency.

3.3. Reductionist framework and need to further tone-down results. Water stress is a crude volumetric indicator of a country's water issues. The authors could consider further toning down their title and abstract to acknowledge that their paper adopts a reductionist approach to study a very complex problem. The abstract goes some way in this direction, but I think it should adopt more cautious language.

Thank you. We wholeheartedly agree that the limitations of the analysis are very important to acknowledge and do so repeatedly in the revised manuscript. For example, the complex nature of water issues that extend beyond a crude measure of water stress are now explicitly discussed (also in response to comment 2.3 from Reviewer 2):

L210 Refugee Resettlement and Food Import

This burden exacerbates a broader landscape of water inequality and injustice in the Global South, where resource mismanagement (Gleick 2014), poor governance (de Chatel 2014, Lu et al, 2014), the commodification of water and land (d'Odorico et al, 2017), extractive industries (Chiarelli et al, 2022), environmental cost shifting (Dell'Angelo et al, 2018) and the legacies of colonialism (Gasteyer et al 2012) often amplify water scarcity in arid and semi-arid regions that will be strongly impacted by climate change (Falkenmark et al, 2013). We recognize that mitigating these issues is a complex undertaking that requires appreciation for the 'relational, situated, and context-sensitive rather than universalistic' nature of water justice (Zwarteveen et al., 2014; Sultana 2018), and that focusing on the water stress of refugee displacements is inevitably a partial -- and somewhat reductionist -- perspective. With these caveats in mind, we investigate the potential to alleviate this increased water burden,} either by transferring virtual water demand *\emph{away}* from vulnerable host countries using refugee resettlement programs; or by transferring virtual water supply *\emph{towards}* vulnerable countries through enhanced food imports.

We also explicitly caution against using water considerations as a sole basis for policy (also in response to your comment 3.8 below):

L261: Policy Implications (note: Bold Emphasis Added)

Global refugee displacement is a complex multi-dimensional crisis that is upending the lives of millions of real humans. Of course, focusing on water-related impacts (even when accounting for "refugee hardship", as in Figure 3B) cannot fully capture the myriad of challenges faced by both the refugees themselves, and by their countries of refuge. To be clear, **we are not suggesting that water stress should be singled out as a basis for trade, aid and migration policy**, which is driven by a range of social, economic, cultural, political and ethical factors that have nothing to do with water (Alix-Garcia et al, 2017). Similarly, the objective of water management should not necessarily be to minimize use of water resources, but rather to ensure that the value of water for society and the environment is maximized (Wichelns 2004). **Rather, our analysis**

positions itself within the framework of the Sustainable Development Goals, where water stress reduction (SDG indicator 6.4.2) is one of many indicators serving to monitor progress towards a broader set of sectoral goals in a way that embraces (or at least acknowledges) their complex interactions.

Unfortunately, word limit considerations restrict our ability to alter the title and abstract much. We have not altered the title, which we believe is a neutral description of our analysis (as it should be). We have adjusted the last sentence of the abstract as:

Abstract

While water considerations should not, alone, determine trade and migration policy, we find that small changes to current international food supply fluxes and refugee resettlement procedures can **potentially** ease the effect of refugee displacement on water stress in vulnerable countries.

3.4. Explain the novelty of these findings to Jordanian policy-makers. A reader familiar with the Middle East and Jordan will not be surprised to find out that millions of refugees exacerbated water issues in Jordan, which already had severe water issues even before the Syrian crisis (e.g., non-revenue water, a highly precarious financial situation of the water sector due to escalating electricity costs). While the authors note this in some parts of the manuscript, this should be front and centre. The authors are calculating the blue water footprint of food demand from refugees – this is elegant and gives a new quantitative estimate but does not really provide any new policy insight to Jordanian policy-makers in my opinion. A recent policy report from the Ministry of Water and Irrigation¹ came to the same conclusion but at least provided a much more balanced view of reality on the ground and considered a wider range of policy options.

Our findings point to Jordan as a critical hotspot for water stress related to refugee displacements, so we agree that emphasizing their practical implication for that location is important. Unfortunately, the global scope of the paper limits our ability to delve into the issue in a way that would do it justice. With this as a caveat, we have extended our discussion about Jordan with explicit reference to the relevant literature. This includes the report that you suggest, which focuses on the pressure imposed by refugee displacement on water supply infrastructure and water delivery services. This paper's contribution is different in that we focus on the water associated with the production and import of the food consumed by the refugees. In particular, we account for likely differences in diets compared to the local population, which likely gives rise to a different per-capita footprint in the short run. This is a new dimension of the water-refugee conundrum in Jordan that has not been directly estimated in our knowledge (also see our response to comment 1.3 from reviewer 1, above). While we strongly caution against using them directly to inform policy (e.g., L263), we do believe that our quantitative estimates are important to inform more elaborate models of Jordan's water sector (e.g., Ref 10), which have undeniable policy value. Finally, as elaborated in response to your second comment (comment 3.2), our results also clarify the water balance of changing

flow regimes in the Yarmouk, which is a recurring hot topic in regional water policy (see, e.g., Ref 47). Our revised discussion about the Jordanian water sector reads:

L175 (Regional Impacts)

The multi-faceted challenges faced by the Jordanian water sector are well documented (Yoon et al, 2021, Syhns et al 2015, Hussein et al 2017, Courcier et al 2005, Whitman et al 2019, Mustafa et al 2019, Al Sabeh et al, 2022), and include the need to supply water to an increasing population of refugees (Breulmann et al, 2021). Unlike previous efforts, this study documents the **emph{food}** demand displaced by refugee displacements into Jordan, which we estimate has contributed 47 to 75 percentage points to the country's overall water stress (Table 2, JOR short and long run).

Our discussion on the practical contribution of our finding reads

L 272 Policy implications

We also believe that the first quantitative estimate of virtual water fluxes associated with forced population displacement that we provide is helpful to inform more complex operational models to support policy (see e.g., Yoon et al, 2021 for Jordan).

3.5 Policy alternatives and decision to focus on refugee resettlement policies. Tony Allan came up with the notion of virtual water trade to explain how countries in the Middle East continue to grow economies and populations despite water scarcity (Allan 2012) [Please cite Tony Allan – without his contribution we would not be discussing this paper – and also have a look at the recent collection of papers published in Water International discussing his contribution].

Thank you -- we have added the suggested reference in the manuscript (L 250)

3.6. The authors use this framing to consider virtual water trade as one of the policy levers that Jordan can deploy to deal with refugee-linked water scarcity. This is good. I suggest stopping there so as to remain within the narrow 'volumetric' framing of the paper and not enter refugee resettlement policy. If the authors want to keep the refugee resettlement policy trade-off analysis, then they should at least explain why they single it out among additional policy alternatives. This might include, for example, investments in hydroponic farming to give refugees the opportunity to grow their own food without consuming water and land resources (Verner et al. 2017) This is just an example of the wide portfolio of options not considered in the paper. I suggest the policy analysis section to just focus on virtual water calculations.

We agree that resettlement or repatriation is the least practical option for the moment. (Note that our results for resettlement are equally valid for repatriation because they only consider the current country of refuge.) However, we do believe that these outcomes are the most durable alternatives to address the refugee question in the long run. We think that this should be kept as a long term aspirational goal and hence kept it in the paper. We make sure to make the challenges associated with resettlement -- and its checkered

historical success -- clearer in the revised manuscript and also mention alternative shorter term strategies (such as the one you suggest) and trade as more realistic shorter term approaches. The broader point we are making is that a bundle of alternatives are likely necessary to address this issue in different contexts and at different time scales. Our revised policy analysis section reads:

L275 (Policy Implications, with relevant changes in bold)

Reassuringly, our results have shown that water stress associated with increased food consumption is not a prominent issue in the overwhelming majority of destination countries. {This finding is important in the context of increasing rhetoric from the far-right that present migrants and refugees as drains the resources of host countries (see, e.g., (Beck et al, 2017) for the European Union). In a few specific countries, however, the added water demand associated with the food consumed by refugees has the potential to destabilize an already overextended water sector. **{Along with interventions to improve the water use efficiency of local food production (e.g., Verner et al, 2017)}**, enhanced import of water-intensive foods from water abundant countries, either through food aid or favorable trade conditions, stand out as promising approaches to mitigate impending water crises in the short run. Yet international food supply chains are themselves vulnerable to conflict-related shocks, as illustrated by the ongoing crisis in Ukraine. A resilient trade network, where countries can respond to shocks by forming new partnerships, is essential to avert the cascading effects of conflicts on food security. Such networks are facilitated by policies that support the diversification of import sources in vulnerable countries (Karakoc et al., 2021). **In the long run, the safe repatriation of refugees in their country of origin -- or their long term relocation in countries where they can sustainably build a new life -- stand out as important aspirational goals.** Refugee resettlement can simultaneously provide a lasting solution for the refugees themselves, while relieving the often resource-poor countries that currently shelter them. Our analysis illustrates the dynamics of this win-win proposition in the specific context of water stress. Yet, as of 2021, less than 3\% of the 1.44 million refugees of the UNHCR plan have been successfully relocated (UN 2021). There is a salient need -- and a moral imperative -- for higher-income countries to support the resettlement effort through increased resettlement pledges.

3.7. Implicit assumption that water released by resettlement will not be used in Jordan. By suggesting that refugees hosted in Jordan should be resettled somewhere else to relieve water stress, the paper assumes that the water 'released' by resettling refugees will not be used by other water users in Jordan. The authors should clarify if they are implicitly assuming that the water released by resettling refugees outside of Jordan would not be used to meet other uses in Jordan (thus negating any reductions in water stress).

This is an excellent point that prompted us to add an important robustness check to our analysis. By assuming that the water 'released' by the resettled refugees will fully serve to relieve water stress, we implicitly assume that per capita water consumption is not affected by water availability. This is, indeed, a strong assumption that calls for a

dedicated robustness check. In a new analysis in SI, we drop this assumption and instead assume that the per capita water footprints increase linearly with per capita water availability -- an assumption that is consistent with available data in many countries (Figure S8 A and Table S8). We found that allowing for per capita water use to be dependent on per capita water availability does not materially affect the results of our original analysis in Figure 3B. We report these new results in the Methods section, referring to the relevant SI text and figure:

L 423 (Methods)

This assumption implies that the water 'released' by resettling refugees will fully serve to relieve water stress in their former country of refuge, and not instead be used by other water users. In other words, we assume that the people remaining in the country will not increase their consumption in response to the increased water availability associated with the resettlement. This implies in particular that the local per capita water footprint is independent from water availability. We relax this assumption in a robustness check presented in SI, where we instead assume that the per capita water footprints increase linearly with per capita water availability -- an assumption that is consistent with available data in many countries (Figure S8 A and Table S8). The outcomes of this new analysis (Figure S7 B) are materially similar to our results in Figure 3b.

We have also added a full page to the SI that describes how our original approach is expanded to account for the new assumption and a new SI figure to present the results.

3.8. Implications for migration policy more in general. I think it is useful to take a step back and consider the paper's implications for migration policy more in general. In other words, let's consider the paper's implications for economic migrants, who account for a much larger share of overall population movement. According to the paper's framing (if I understand things correctly), population movement transfers blue water demand towards countries with water-intensive dietary habits. This effect is obviously more pronounced in countries with already high blue water stress. Hence, if we blindly apply the paper's approach, we should suggest to Qatar and UAE (two countries with high levels of freshwater stress) to reduce their very large populations of economic migrants because they increase blue water stress. I think we would tend to agree that such policy recommendation would be misplaced, not least because it ignores the migrants' contribution to these countries' economic productivity and wealth. Can the authors comment on this point, which in my opinion demonstrates the rather reductionist approach taken in the paper?

We do not suggest that water stress consideration should determine economic migration in arid regions (although recent development in the Colorado River basin suggest that it would perhaps be wise to consider it as a constraint to long term development). We agree that basing migration policy (or any policy decision) on the single objective of reducing water stress is an extremely reductionist approach and make it clear throughout the manuscript (e.g., Abstract and L264) that this is not what we suggest.

Rather, our analysis positions itself within the framework of the sustainable development goals, where water stress reduction (SDG 6.2.1) is one of many indicators serving to monitor progress towards a broader set of sectoral goals in a way that embraces (or at least acknowledges) their complex interactions. We have added language in the revised manuscript to make the connection to the SDGs and the associated cross-sectoral paradigm more explicit. Our policy section now includes (with new text in bold):

L263 Policy Implications

To be clear, **we are not suggesting that water stress should be singled out as a basis for trade, aid and migration policy**, which is driven by a range of social, economic, cultural, political and ethical factors that have nothing to do with water (Alix-Garcia et al, 2017). Similarly, the objective of water management should not necessarily be to minimize use of water resources, but rather to ensure that the value of water for society and the environment is maximized (Wichelns 2004). **Rather, our analysis positions itself within the framework of the Sustainable Development Goals, where water stress reduction (SDG indicator 6.4.2) is one of many indicators serving to monitor progress towards a broader set of sectoral goals in a way that embraces (or at least acknowledges) their complex interactions.**

3.9. MINOR POINTS • Line 74-75: this seems to be factually incorrect given current refugee numbers : Venezuela, Ukraine and South Sudan are among the top 5 countries of origin of refugees, and they are all water rich countries (and so are the host countries). Please consider revising this statement and providing the years it refers to.

Agreed, we removed “and lower than average” from that sentence:

L81. Global Footprint

In contrast, most refugee fluxes occur locally and connect countries with comparable per capita water footprints.

3.10 CONCLUDING REMARKS Please note again that I am very impressed by the paper’s ambition and rigorous analytical approach. This work is of high quality. However, having worked on migration issues myself, I just want to really advise the authors to keep the paper on a defined, narrow track and be as humble as possible in presenting findings.

Thank you very much -- we appreciate your input and hope that our responses and revised manuscript address your concerns.

References

- Allan, T. (2012). The Middle East water question: Hydropolitics and the global economy. Bloomsbury Publishing.
- Verner, D., Vellani, S., Klausen, A. L., & Tebaldi, E. (2017). Frontier Agriculture for Improving Refugee Livelihoods. World Bank: Washington, D.C.